# Telomeres reforged with non-telomeric sequences in mouse embryonic stem cells

Chuna Kim[1,2,7], Sanghyun Sung[1,7], Jong-Seo Kim [1,3,7], Hyunji Lee[1], Yoonseok Jung[3], Sanghee Shin[1,3], Eunkyeong Kim[1], Jenny J. Seo[1,3], Jun Kim [1], Daeun Kim[4,5], Hiroyuki Niida[6], V. Narry Kim [1,3], Daechan Park[4,5 ✉] & Junho Lee[1 ✉]

Telomeres are part of a highly refined system for maintaining the stability of linear chromosomes. Most telomeres rely on simple repetitive sequences and telomerase enzymes to protect chromosomal ends; however, in some species or telomerase-defective situations, an alternative lengthening of telomeres (ALT) mechanism is used. ALT mainly utilises recombination-based replication mechanisms and the constituents of ALT-based telomeres vary depending on models. Here we show that mouse telomeres can exploit non-telomeric, unique sequences in addition to telomeric repeats. We establish that a specific subtelomeric element, the mouse template for ALT (mTALT), is used for repairing telomeric DNA damage as well as for composing portions of telomeres in ALT-dependent mouse embryonic stem cells. Epigenomic and proteomic analyses before and after ALT activation reveal a high level of non-coding mTALT transcripts despite the heterochromatic nature of mTALT-based telomeres. After ALT activation, the increased HMGN1, a non-histone chromosomal protein, contributes to the maintenance of telomere stability by regulating telomeric transcription. These findings provide a molecular basis to study the evolution of new structures in telomeres.

[1] Department of Biological Sciences, Seoul National University, Seoul, Korea. [2] Aging Research Center, Korea Research Institute of Bioscience and Biotechnology, Daejeon, Korea. [3] Center for RNA Research, Institute for Basic Science, Seoul, Korea. [4] Department of Biological Sciences, Ajou University, Suwon, Korea. [5] Department of Molecular Science and Technology, Ajou University, Suwon, Korea. [6] Department of Molecular Biology, Hamamatsu University School of Medicine, Hamamatsu, Shizuoka, Japan. [7] These authors contributed equally: Chuna Kim, Sanghyun Sung, Jong-Seo Kim. ✉email: dpark@ajou.ac.kr; elegans@snu.ac.kr

Ends of linear chromosomes should handle two problems: 'the end replication problem' in which DNA replication machinery cannot completely replicate ends of lagging strands, and 'the end protection problem' in which chromosomal ends should be discriminated from internal double-strand breaks (DSBs)[1,2]. Telomeres are nucleoprotein structures that solve these problems and are usually based on simple repetitive sequences. If cells continue to divide without a mechanism to maintain telomere length, telomeres gradually shorten leading to loss of protective function, the recognition of chromosomal ends as breaks, and the activation of the DNA damage response (DDR). Affected cells usually enter cellular senescence, an irreversible cell cycle arrest[3,4]. Therefore, a telomere maintenance mechanism is important for cellular proliferation which is linked to species permanence, stem cell maintenance and tumorigenesis. The majority of cells in eukaryotic organisms, including human cancer cells, maintain telomere length by using the telomerase enzyme, but ~15% of human cancer cells are known to maintain telomeres by telomerase-independent maintenance mechanisms, which are summarised under the term alternative lengthening of telomeres (ALT)[5–7].

In addition to gradual shortening due to cell division, telomeres can be damaged by various cellular stresses, such as replication stress and oxidative stress due to their inherent structural characteristics. In particular, stochastic telomere deletions (STDs), which generate large or complete telomeric deletions over a short period, appear to occur when telomeres are unable to handle DSBs caused by internal and external stresses[8]. DSBs within telomeres can be resolved by telomerase. However, in cases where telomerase cannot repair DSBs, STDs may lead to telomeric rearrangements mediated by ALT-like recombination or chromosomal fusions, which can cause genome instability. Degenerated telomeres and segmental duplications, which may be the remnants of previous telomeric repairs, are enriched within subtelomeres, the telomere-proximal regions. Genomic instability caused by telomere dysfunction is mostly catastrophic, but in certain cases, it can be adaptive and lead to rapid genomic evolution. ALT may be a mechanism promoting chromosomal diversity in the course of handling frequent telomere crisis.

The maintenance mechanisms of telomeres have been developed through evolutionary pathways to ensure chromosome integrity. Given that telomerase is widespread in eukaryotes, telomerase-based telomere maintenance appears to have developed early in the evolution of eukaryotes. However, in certain species that have lost the telomerase gene, ALT is the sole telomere maintenance mechanism. For instance, *Drosophila melanogaster* naturally uses retrotransposons and *Allium cepa* contains satellite and rDNA repeats as telomeric sequences[9,10]. Experimental telomerase deletion can also lead to alternative telomere formation in some species: *Saccharomyces cerevisiae* type I survivors[11], *S. pombe* heterochromatin amplification-mediated and telomerase-independent (HAATI) survivors[12] and *C. elegans* TALT (template for ALT) survivors[13]. Telomeres of human ALT cancers seem to consist only of telomeric repeats and their variants. Interestingly, there is a distinct human cell line, AG11395, which is a SV40-transformed Werner mutant fibroblast. Telomeres of AG11395 cells contain extensive amounts of SV40 sequences and telomeric repeats[14,15]. The cells showed some typical ALT characteristics, but lacked ALT-associated promyelocytic leukaemia bodies (APBs), a molecular marker of ALT. This case implies the possibility that ALT may be a multifaceted mechanism.

Previous studies have reported that non-telomeric sequences were located in the telomeres of mouse embryonic stem cells (mESCs). Terc (telomerase RNA component) knock-out mESCs continue to proliferate with shortened telomeres and eventually cease to divide due to telomere-induced senescence[16]. However, a small fraction of cells overcame the telomere crisis and became immortalised again (Fig. 1a, Supplementary Fig. 1a). Notably, one line of the ALT survivor cells (DKO 741, hereafter named the 'ALT mESCs') contained non-telomeric sequences in its telomeric region. In light of this finding, it is conceivable that mammalian telomeres may also be reconstructed. However, it is unknown where the non-telomeric sequences originate from, how they are maintained and what unique characteristics the ALT cells have. Fortunately, ALT mESCs were stored chronologically providing an opportunity to compare the various features of the cells before and after ALT activation.

In this study, we found that the sequence used as a template for ALT telomeres (named as mTALT) originated from a specific subtelomeric region. After cells overcame cellular senescence, they contained a new telomeric structure in which the mTALT unit and telomeric repeats coexisted. During the stabilisation of the ALT mechanism, the expression of HMGN1 protein increased. HMGN1 contributed to telomere maintenance by increasing the expression of telomere repeat-containing RNA (TERRA) in ALT telomeres. Our findings suggest that non-telomeric sequences from the genome can be parts of telomeres even in mammalian cells. The evolutionary conservation of this ALT phenomenon implies that eukaryotes have a robust system to cope with the loss or inactivation of telomerase.

## Results

**ALT mESCs amplify the mTALT sequence to maintain telomere length.** First, we examined the growth rate of the ALT mESCs at several time points: population doubling 100 (PD100), at which pre-ALT cells are almost normal due to the remaining long telomeres; PD350, at which pre-ALT cells are approaching the expected senescence point; PD450, at which post-ALT cells have just escaped from senescence, and PD800, at which post-ALT cells are stably surviving. Compared with the PD100 cells, PD350 cells showed a significantly slower growth rate (Fig. 1b). After PD450, the cell growth rate recovered without telomerase, indicating that ALT was activated before PD450 and maintained thereafter. Molecular markers of ALT cancers, such as variant telomeric repeats (e.g., TCAGGG and TGAGGG), C-circle (extrachromosomal telomeric DNA circle of C-rich strand), and ALT-associated promyelocytic leukaemia bodies (APBs), were not different between pre-ALT and post-ALT cells (Supplementary Fig. 1b–e)[6,17], increasing the likelihood that ALT mESCs have different characteristics from known ALT cancer models.

Quantification of the canonical telomere repeat sequence, TTAGGG, using whole-genome sequencing (WGS) data showed that telomere length decreased from PD100 to PD350 and gradually increased after ALT activation (Fig. 1c). However, telomere constituents changed after ALT activation. A terminal restriction fragment (TRF) assay revealed that only post-ALT cells showed discrete bands with *Alu*I, *Mbo*I, and *Hinf*I which are not capable of restricting the canonical telomeric repeat sequence (Fig. 1d). Additionally, non-telomeric sequences were identified at significantly higher levels within paired reads of telomeric sequence-containing reads after ALT activation (PD450) and with additional passages (PD800) (Fig. 1e). These data suggest that non-telomeric sequences are present within or next to the telomeric repeats in post-ALT cells.

To deduce ALT-specific telomeric constituents, we investigated the location of the non-telomeric sequences using WGS data. The terminal region of the long arm part of chromosome 13 was significantly amplified in post-ALT cells (Fig. 1f). The mapped reads on the reference genome showed a specifically amplified genomic region spanning ~7.4 kb in post-ALT cells (Fig. 1g). We

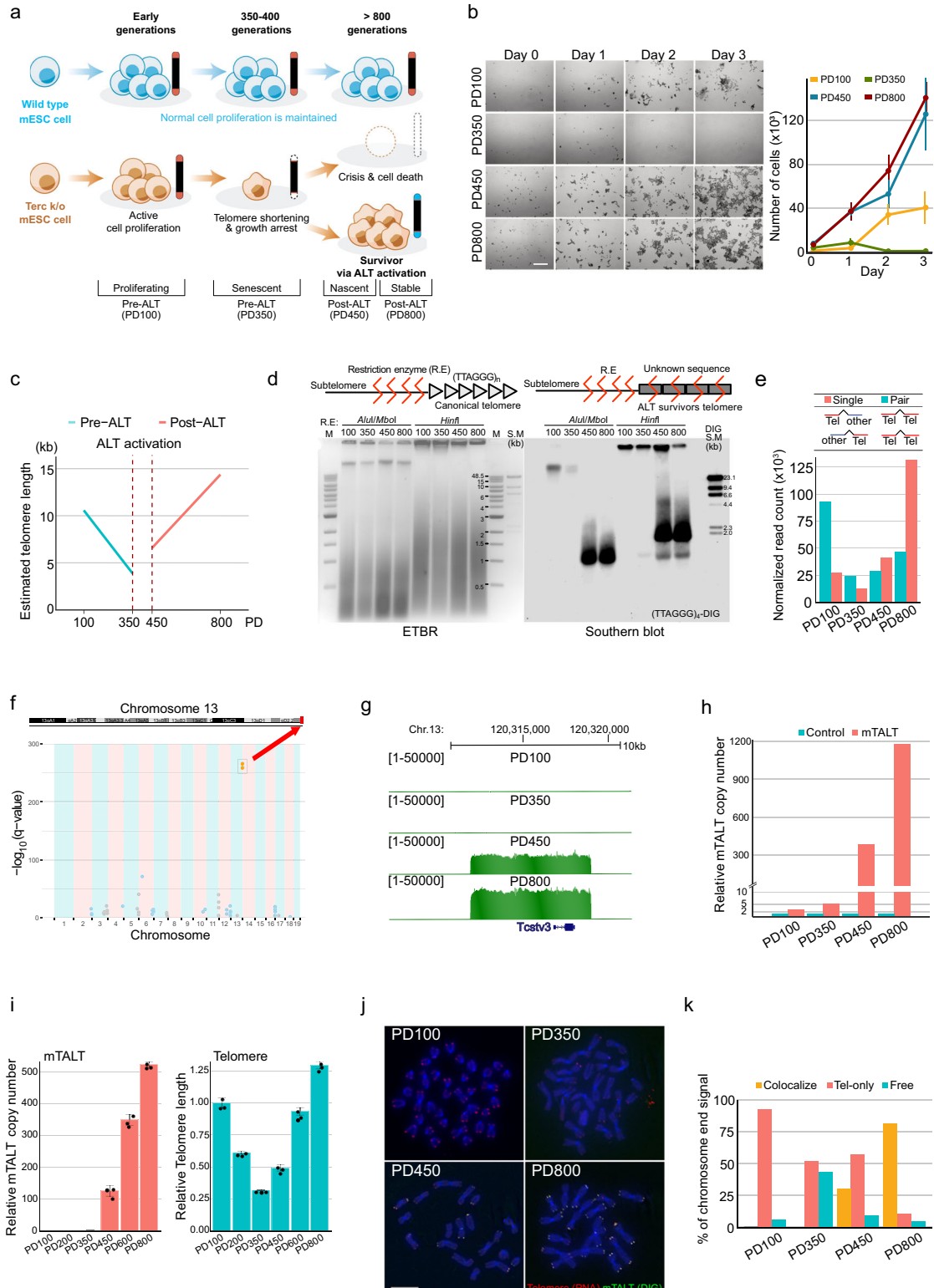

named the region *mouse template for ALT* (mTALT). The copy number of mTALT increased as cells progressed from the pre-ALT state to ALT activation and stabilisation (Fig. 1h, i, Supplementary Fig. 1f, g). Using a PCR assay, mTALTs were also detected at locations other than the original site at chromosome 13 (Supplementary Fig. 1h).

While ~30% of telomeres of the PD450 cells colocalised with mTALT, telomeres of PD800 cells showed ~80% of co-localisation,

indicating that the ALT telomeres were stabilised even with partial recruitment of mTALTs in early post-ALT cells (Fig. 1j, k). After ALT activation, telomere dysfunction-induced foci (TIF) were increased and maintained until PD800 (Supplementary Fig. 1i, j). The increased TIF may reflect the propensity of ALT cells to spontaneously form DSBs in telomeres[18]. Taken together, we concluded that ALT mESCs amplify the mTALT sequence to maintain telomere length.

**Fig. 1 mESCs amplify the mouse template for ALT (mTALT) sequence to maintain telomere length. a** Schematics of retrieving ALT mESCs. **b** Growth rate assay with crystal violet staining. (left) Representative images of each cell at daily intervals. (right) Quantified results. The bars represent means and SDs from ≥20 images per condition over five independent experiments. Scale bar, 20 μm. **c** Telomere length quantified from WGS. **d** TRF assay with DIG-(CCCTAA)*4 probe. Genomic DNA samples were cut using the indicated enzymes. S.M size marker. **e** Telomere length quantified from WGS according to read types. 'Pair' indicates that both read mates contain telomeric repeats, and 'Single' indicates that one read does. **f** Mapping result of the mate reads of single telomeric reads. **g** Snapshot of mTALT region with the mapped WGS data. **h** Calculated copy number of mTALT from WGS. Control, chr13:109,998,778-110,006,148. **i** Calculated copy number of mTALT from mmqPCR. mTALT-specific or telomere-specific protocols were used. The bars represent the means and SDs from three biologically independent replicates. **j** Representative images of fluorescence in situ hybridisation (FISH) of mTALT and telomeric repeats. Scale bar, 5 μm. **k** Quantification of mTALT and telomere co-localisation. ≥16 cells per condition were taken. Source data are provided as a Source Data file.

**An mTALT duplicated before ALT activation is selected as the template for new telomeres**. Both telomeric repeats and mTALT sequences are amplified after ALT activation in mESC. To examine how mTALT constitutes new telomeres at a sequence level, we analysed telomere-adjacent sequences by using paired-end WGS reads. Alignment of matched reads where one end contains telomere repeats were exclusively enriched at both the 5′ and 3′ ends of mTALTs (Fig. 2a, left), suggesting that telomere repeats border the terminal regions of mTALTs. The matched reads where one end contains telomere repeats of a specific G- (leading) or C-rich (lagging) strand mapped onto only one end of mTALT (Fig. 2a, right), indicating that mTALT sequences are flanked by telomeric repeats with a uniform directionality. A PCR assay able to distinguish the direction of connected mTALTs showed that mTALTs were duplicated in a relatively regular head-to-tail pattern (Fig. 2b). Interestingly, PD350 cells also showed a PCR band of head-to-tail pattern, indicating that a *cis*-duplication event of mTALT occurred before ALT activation.

Unexpectedly, the PD100 cell contained two copies of mTALT (Fig. 1h, Supplementary Fig. 1f) unlike the reference mouse genome (C57BL/6J) which contains a single copy of mTALT on chromosome 13. Analysis of the WGS data from various mouse strains revealed that most strains contained a single copy of mTALT, but the 129/Ola strain, the parental strain of DKO 741 ALT mESCs, contains two copies while the SPRET strain has about five copies of mTALT (Fig. 2c). This implies that an increase in mTALT copy number in PD100 cells occurred before telomere shortening. To determine whether the two copies of mTALT in the 129/Ola strain were duplicated side-by-side or at different chromosomal locations, we analysed the PD100 WGS data and found that the second copy of mTALT was located at the end of chromosome 11 (Supplementary Fig. 2a, e, f, refer to Supplementary Fig. 3 for a complete sequence of mTALT on chromosome 11). The mTALT on chromosome 11 was located after eight copies of telomeric repeats joined to the subtelomeric region with ~4.5 kb deletion which did not exist in the C57BL/6J strain. Chromosome 11 specific rearrangements such as sub-telomeric deletion, addition of short telomeric repeats and *trans*-duplication of mTALT may have occurred in the process of repairing DNA damage in ancient 129/Ola strains. A PCR assay confirmed the association of chromosome 11 and mTALT (Fig. 2d, e, Supplementary Fig. 2b). The in silico digestion of the mTALT sequence confirmed the outcome of the TRF assay (Supplementary Fig. 2c). When the mTALT of chromosome 11 was cloned and sequenced, the region was found to be homozygous (Supplementary Fig. 2d).

Next, we investigated how accurately mTALTs were replicated in telomeres. When we tracked the single nucleotide polymorphisms (SNPs) in the mTALT region by time points, PD800-specific or post-ALT-specific SNPs were extremely rare (Fig. 2f, Supplementary Fig. 2g). This indicates that the replication mechanism of mTALTs is relatively accurate. Additionally, mTALT had a large number of heterozygous SNPs in pre-ALT and more homozygous

SNPs after ALT activation (Fig. 2g, Supplementary Fig. 2h). Homozygous SNPs in amplified mTALTs originated from the mTALT on chromosome 11, not 13, suggesting that the mTALT on chromosome 11 was selectively amplified and used for terminal maintenance in post-ALT cells (Fig. 2h). New ALT survivor cells obtained from independent experiments also duplicated the mTALT of chromosome 11 (Fig. 2i, Supplementary Fig. 2i).

As the cells progressed from PD100 to PD350, the copy number of mTALT increased, and the mTALT of chromosome 11 was selectively amplified (Supplementary Fig. 2j, k). We concluded that mTALT that was duplicated independently of ALT activation was selected as the template for newly formed ALT telomeres.

**Selected ALT mESCs entail subtelomeric copy-number variations (CNV)**. In human cancers, ALT activation is frequently accompanied by genomic instability and specific mutations such as ATRX, DAXX, or H3.3 mutations[19]. To examine whether genetic mutations accompanied ALT activation in our ALT mESCs, we compared genome-wide SNPs present at each time point (Fig. 3a, Supplementary Fig. 4a). The number of post-ALT-specific SNPs (indicated in blue) was not greater than others, indicating that few mutations occurred during ALT activation. Post-ALT-specific SNPs present in exons do not appear to affect gene expression significantly (Supplementary Fig. 4b). Genome-wide loss-of-heterozygosity (LOH) increased from PD100 to PD350 in terms of SNPs and indels (Supplementary Fig. 4a). LOH can occur during the process of repairing DNA damage using homologous chromosomes rather than sister chromatids. A closer look at the subtelomeric regions of PD350 showed that the ratio of homozygous SNPs in chromosome 11 gradually increased when approaching the end of the chromosome (Fig. 3b), suggesting repair mechanisms that induce LOH may have been active in the subtelomere of chromosome 11 before the start of cellular senescence.

Unlike SNPs, CNVs increased significantly after ALT activation, particularly in the terminal region of chromosomes. When compared with the PD100 genome, genomic aberrations in PD450 and PD800 cells were observed at the long arm subtelomeres (Fig. 3c). The amount of various repetitive sequences, including rRNA, did not significantly change (Supplementary Fig. 4c). This suggests that CNVs were enriched in the long arm where telomere lengthening occurred and not in the short arm where telomere fusions mainly occurred. Interestingly, a few CNVs were specific to PD800, i.e., the terminal regions of chromosomes 2, 5, and 16, suggesting that telomeric instability was sustained after ALT activation and led to CNV (Fig. 3c). This is consistent with the increased and persistent DDR in post-ALT cells (Supplementary Fig. 1i, j).

The PD350 genome showed only a few CNV regions except for chromosome 11 (Fig. 3c, d). Various low-fold copy-number

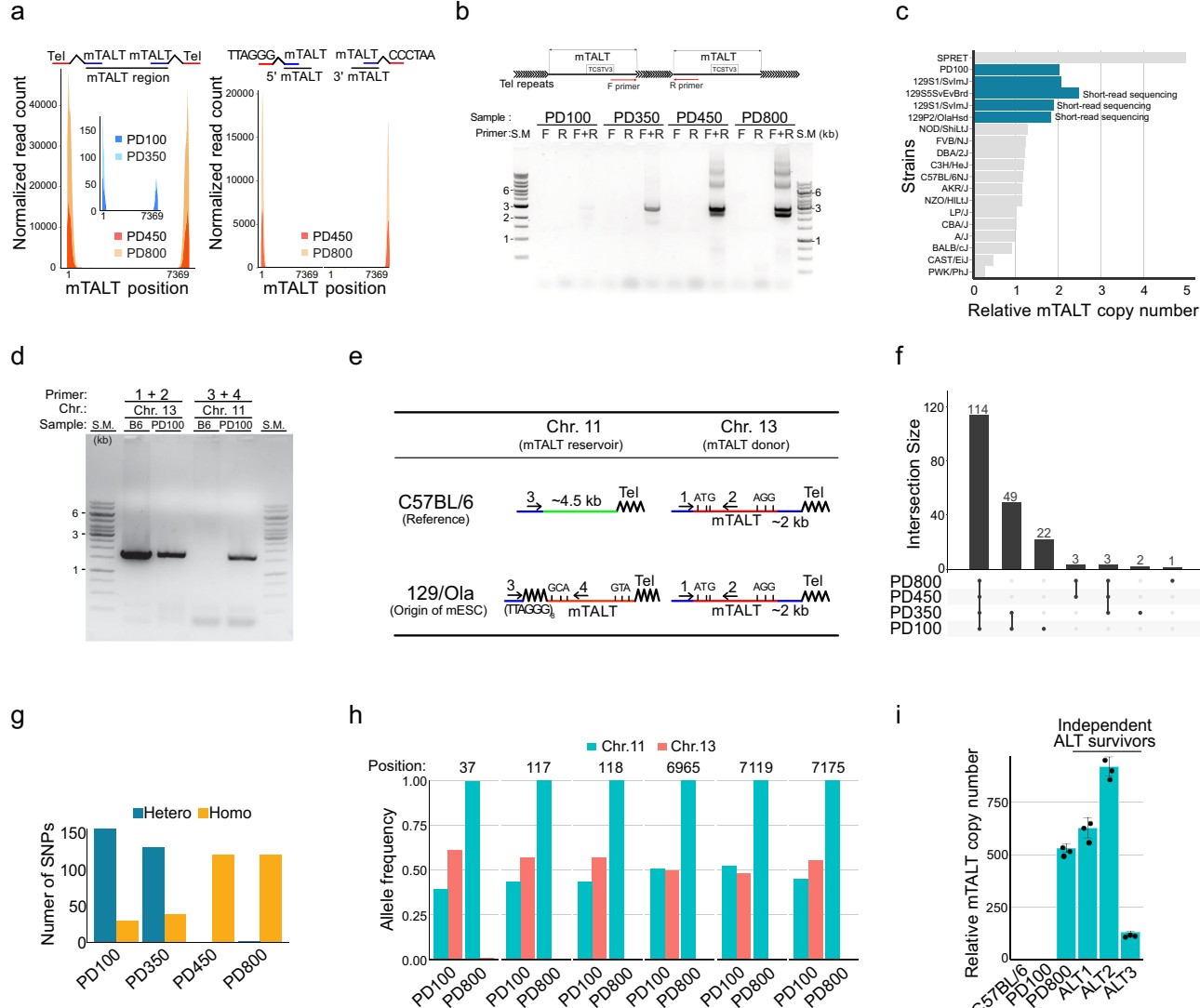

**Fig. 2 A specific copy of mTALT is duplicated in a constant orientation. a** (Left) Alignment of the mate reads of telomere-containing reads. (Right) Alignment of the mate reads of telomere-containing reads in a strand-specific way. **b** (Top) Schematics of the tandemly duplicated mTALTs and the specific primers used. (Bottom) PCR assays showing the directionality of mTALT duplication. **c** Calculated copy numbers of mTALTs from WGS data of various mouse strains. **d** PCR assays elucidating the presence of mTALT on chromosome 11. **e** Schematics of the mTALT region in chromosomes 11 and 13 of the reference (C57BL/6J) and 129/Ola genome. **f** SNP analysis of the mTALT region at each time point. The connected lines denote groups sharing SNPs. **g** Number of SNPs sorted by allelic zygosity. **h** Frequencies of representative SNPs in mTALT region. **i** Calculated copy numbers of mTALTs from independent ALT survivors. The bars represent the means and SDs from three biologically independent replicates. Source data are provided as a Source Data file.

changes on chromosome 11 may indicate that individual cells went through various rearrangements and thus comprised a heterogeneous population. During the transition to PD450 cells, the complex pattern of CNVs in this region mostly disappeared and remained only at the subtelomeric regions, indicating that certain cells or a single cell were selected as they activated ALT (Fig. 3d).

**Long non-coding RNA is important for telomere maintenance in ALT mESCs.** We next examined epigenetic changes in ALT cells by performing an assay for transposase-accessible chromatin using sequencing (ATAC-seq) analyses[20]. Unsupervised clustering analysis using ATAC-seq peak intensities differentiated pre-ALTs from post-ALTs (Fig. 3e, f, Supplementary Fig. 4d–f). When the differential peaks were classified into functional terms,

the insulator-related and enhancer-related terms showed the most difference (Fig. 3g, h, Supplementary Fig. 4g).

To determine the epigenetic state of the mTALT-containing ALT telomeres, we quantified ATAC-seq peaks for telomeric repeats and mTALT sequences. Telomeric repeats were more epigenetically open in PD350 cells than in PD100 cells and then gradually became less open after ALT activation (Fig. 4a). mTALT sequences also became epigenetically less open after ALT activation (Fig. 4a, Supplementary Fig. 4h). When aligning ATAC-seq peaks within the mTALT region, the highest peak corresponded to a CTCF-binding insulator region with extensive opening during ALT activation and stabilisation (Fig. 4b, c).

As previously reported, CTCF-driven telomeric repeat-containing RNA (TERRA) transcription can facilitate telomeric DNA replication and chromosome stability[21]. Since the insulator region of mTALT has more accessible chromatin than other

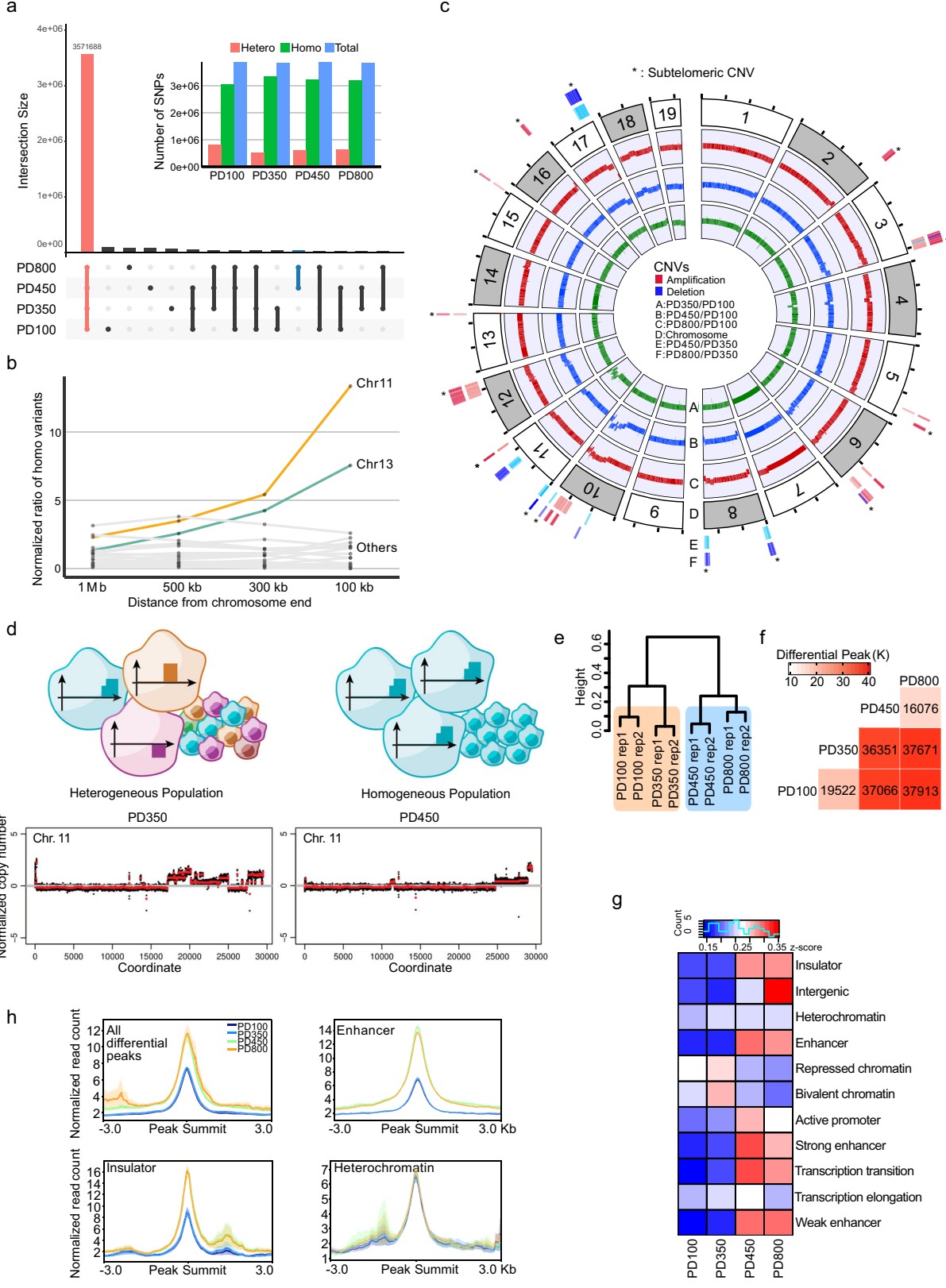

telomeric regions, we speculated whether mTALT could produce non-coding RNA which could affect telomere maintenance. We examined RNA-seq data and found that transcripts including mTALT and TERRA were produced and that they were increased in post-ALT cells (Fig. 4d, Supplementary Fig. 6f). Furthermore, there was a specific direction of transcription to the transcripts: from terminal to internal regions (Fig. 4e). TERRA had chromosome 11-type SNPs that were homozygous. Therefore, TERRA was produced only from the mTALT of chromosome 11 (Fig. 4f). Moreover, long-read Iso-Seq showed that the 5′ end of three mTALT transcripts out of five mapped onto the insulator site (Fig. 4g).

**Fig. 3 ALT mESCs experienced extensive copy-number variations (CNV) and epigenetic remodelling. a** Genome-wide SNP analysis at each time point. The connected lines denote groups sharing SNPs. (Inside) Number of SNPs sorted by allelic zygosity. **b** Numbers of homozygous SNPs of each chromosome focusing on subtelomeric regions. **c** Copy-number variation (CNV) analysis. Green, blue, and red each denote the CNV of PD350, PD450, and PD800, respectively, compared to PD100 as the control. The outermost colour represents the CNV of PD800 compared to PD350, and the inner colour represents the CNV of PD450 compared to PD100. **d** (Top) CNVs of chromosome 11 at PD350 and PD450. (Bottom) Schematics of the cellular heterogeneity at PD350 and the selected result at PD450. **e** Hierarchical clustering of ATAC-seq peaks. **f** Heat map of differential ATAC-seq peaks. **g** Heat map of the functional terms of ATAC-seq data. Colour index indicates the relative proportion of each time point. **h** Normalised differential ATAC-seq peaks aligned to the peak summit. The solid lines and the error bands represent the mean of each data and the standard errors of the mean, respectively. Source data are provided as a Source Data file.

During transcription, the nascent RNA transcript can bind template DNA by displacing non-template ssDNA and form a three-stranded RNA–DNA hybrid (the R-loop)[22]. In addition to the transcription process, RNA generated in one region may *trans*-interact with DNA in another region. R-loops are involved in various physiological processes such as gene regulation and DNA repair, and can also be a cause of DNA damage[23]. To assess whether the telomeric transcripts interact with the transcribed telomeric regions, we conducted a DNA–RNA hybrid immuno-precipitation (DRIP) assay. Post-ALT cells exhibited higher DRIP signals than pre-ALT cells in telomeres (Fig. 4h, i). Down-regulation of TERRA using antisense-locked nucleic acid (LNA) oligos resulted in an increase in TIFs (Fig. 4j, k), indicating that the presence of TERRA promoted a favourable environment for telomere stability in ALT mESCs.

**TRF2 is required for ALT telomere maintenance.** Since canonical telomeric repeats were duplicated along with mTALTs in ALT mESCs, the telomeric repeats and their binding proteins may also retain their protective roles. Shelterin is a protein complex that binds to and protects mammalian telomeres[24]. We examined whether ALT mESCs also depend on shelterin for telomere stability. First, ALT mESCs (PD800) showed TRF2 and telomere co-localisation (Supplementary Fig. 5a). TRF2-depleted PD800 cells died after 2–3 passages (Supplementary Fig. 5b). Before all of the TRF2-depleted cells died, the remaining cells were collected and assessed for TIFs. TRF2-depleted PD800 cells had significantly increased TIFs (Supplementary Fig. 5c, d). Therefore, it can be concluded that telomeric repeats and TRF2 still contribute to the protection of new chromosomal ends of ALT mESCs.

**Identifying HMGN1 as an ALT maintenance factor.** In order to identify the factors regulating TERRA expression and ALT maintenance, we conducted unbiased quantitative proteomics using cells from PD100 and PD800 (Fig. 5a, Supplementary Data 1). The Gene Ontology (GO) terms of the significantly increased proteins in PD800 cells were transcription, chromatin organisation, and stemness (Fig. 5b, Supplementary Fig. 6a–c). We also conducted mRNA sequencing of the same cells (Supplementary Fig. 6d–f, Supplementary Data 2). When aligning the differentially expressed genes from the transcriptome analysis with GO terms from the proteome analysis, we observed that both experiments gave consistent results (Supplementary Fig. 6g).

In a previous report, telomere shortening of mESCs induced the increased expression of stemness genes by genome-wide DNA demethylation, which led to unstable differentiation[25]. Interestingly in ALT mESCs, the increased expression of stemness genes such as *Klf4* and *Nanog* was sustained after ALT activation (Fig. 5a). When these transcription factors were depleted, cell growth slowed and telomere DNA damage increased (Supplementary Fig. 8a, b). Thus, the change in stemness genes may reflect an alteration of transcription networks through changes in transcription factors and chromatin state to maintain telomere

and genome integrity. A different study showed that a portion of mESC populations momentarily show gene expressions similar to those of the two-cell (2C) embryo state, with *Zscan4* at the centre of the regulatory network[26]. 2C embryos and mESCs in 2C states show rapid telomere extension via recombination in a *Zscan4*-dependent manner. The expression of *Zscan4* was low in PD800 cells compared to PD100 cells at RNA and protein levels. This suggests that telomere maintenance in ALT mESCs may be regulated by mechanisms other than the one involving a high level of Zscan4. Meanwhile, human ALT cancers are highly correlated with the inactivation of ATRX and DAXX proteins[19]. Interestingly, the expression of ATRX was increased, rather than decreased, in post-ALT mESCs, and the expression of DAXX was not affected by ALT activation, which implicates a different mode of action of ALT in mESCs.

Next, single-cell RNA-sequencing was performed to determine whether the gene expression changes accompanying ALT resulted from homogeneous or heterogeneous changes across individual cells. PD100 and PD800 cells constituted two distinct populations, and homogeneity in each population was high (Fig. 5c). Upon listing the critical genes that separated the mESCs into two groups by the degree of influence, we found *Hmgn1*, a non-histone chromosomal protein-coding gene (Fig. 5d, Supplementary Fig. 7a, b). HMGN1 was also identified by quantitative proteomics as one of the significantly enhanced proteins, and top-ranked in the chromatin organisation GO term (Fig. 5b, e). HMGN1 is a member of the high mobility group family and is known to compete with the linker histone H1 for binding the nucleosome core particle[27]. HMGN1 plays an important role in controlling chromatin structure and function, leading to changes in DNA metabolism including transcription, replication, and repair[28,29]. *Hmgn1* was uniformly and highly expressed in a post-ALT-specific manner (Fig. 5c, Supplementary Fig. 7c–g). Interestingly, *Hmgn1* is located in the subtelomeric region of chromosome 16, which is amplified in PD800 cells compared to PD450 cells (Fig. 3c, Supplementary Fig. 7h-j). This suggests that the gene duplication of *Hmgn1* may have been coincidentally selected with ALT activity.

Finally, we examined the role of HMGN1 in ALT telomere maintenance. We first assessed the direct interaction between HMGN1 and the telomeres. The co-localisation of HMGN1 and the telomeres was increased in post-ALT cells with immuno-fluorescence (Fig. 5f, g). Additionally, chromatin immunopre-cipitation showed that HMGN1 was enriched in telomeric repeats and mTALT sequences after ALT activation (Fig. 5h). This suggests that HMGN1 binds to telomeres in ALT mESCs. To assess the role of HMGN1 in telomere maintenance, we depleted HMGN1 in post-ALT cells with *Hmgn1*-specific shRNA. HMGN1 knock-down increased TIFs (Fig. 5i). HMGN1-depleted cells that were passaged for about 2 months contained 30% shorter telomeres than control cells (Fig. 5j). The depletion of other HMG genes also increased TIFs but hardly affected telomere length (Supplementary Fig. 8c-e). These data indicate that HMGN1 is necessary for ALT telomere maintenance.

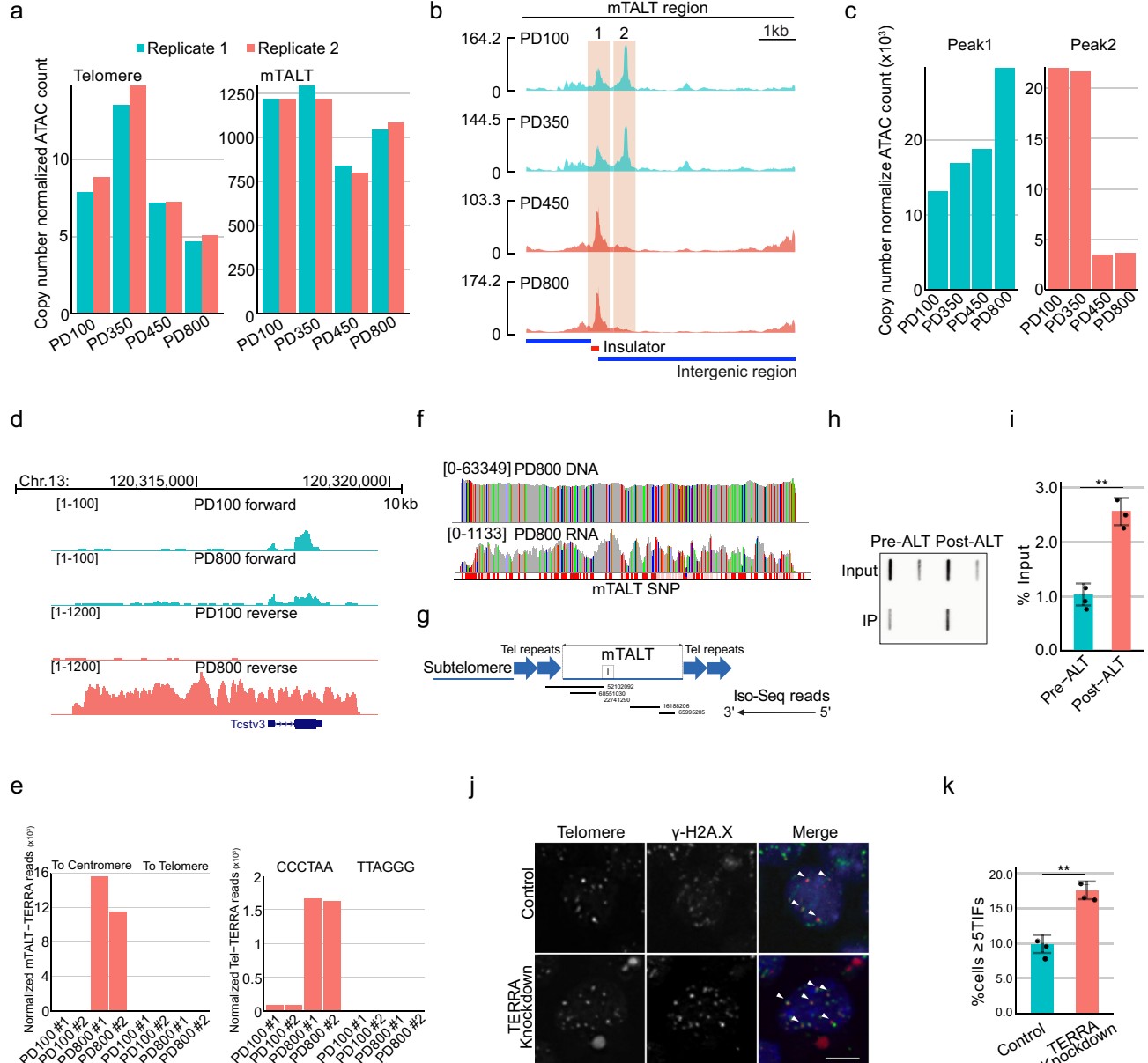

**Fig. 4 Long non-coding RNA transcribed from ALT telomeres is important for telomere maintenance in ALT mESCs. a** Copy number normalised ATAC-seq counts mapped to telomere and mTALT regions. **b** mTALT region snapshot of ATAC-seq peaks. **c** Quantification of ATAC-seq peaks mapped to specific peaks inside mTALT in **b**. **d** Snapshot of RNA-seq of the mTALT region. 'Forward' and 'reverse' denote the transcription directions. **e** Calculated levels of mTALT-containing (left) and telomeric repeat-containing transcripts (right) according to the transcriptional direction. **f** Snapshot of the SNPs of mTALT genomic sequences and transcripts. **g** Iso-Seq reads aligned to the mTALT region. The right ends of all reads are 5′. I insulator. **h** Representative figure of the slot-blot with a telomere-specific probe for the DRIP assay. Input DNA was loaded at a 1/10 dilution. **i** Quantification of slot-blot results. The bars represent the means and SDs from three biologically independent replicates. $P$ value from a two-tailed unpaired $t$-test: **$P = 0.0012$. **j** Representative images of interphase-TIF analysis. Scale bar, 5 μm. **k** Quantification of the ratio of cells with more than five TIFs. The bars represent the means and SDs from three biologically independent replicates. $P$ values from a two-tailed unpaired $t$-test: **$P = 0.0018$. Source data are provided as a Source Data file.

Considering the chromatin remodelling functions of HMGN1, we hypothesised that a high level of HMGN1 might induce telomeric transcription and affect R-loop formation in telomeres. Indeed, mTALT transcripts and telomeric R-loop formation were decreased in HMGN1-depleted cells (Fig. 5k-m, Supplementary Fig. 8f). In HMGN1-depleted cells, the lowered telomeric transcription and telomeric R-loop coincides with higher telomeric DNA damage. To specifically determine the relationship between telomeric R-loop and DNA damage, we depleted RNaseH1, which degrades the RNA moiety of R-loops, and measured TIFs.

Remarkably, RNaseH1 depletion, which increased telomeric R-loops, also induced higher levels of TIFs (Supplementary Fig. 8g). It is reasonable to assume that an elaborate balance of R-loop is important for stable telomere maintenance in ALT mESCs. HMGN1 depletion also decreased telomere fragility which is an indicator of telomeric replication stress (Fig. 5n, o). Taken together, the upregulated HMGN1 in post-ALT cells induced an increase in TERRA and R-loops possibly through changes in local telomeric chromatin structures. R-loop-induced telomere instability may lead to telomere lengthening and chromosome maintenance.

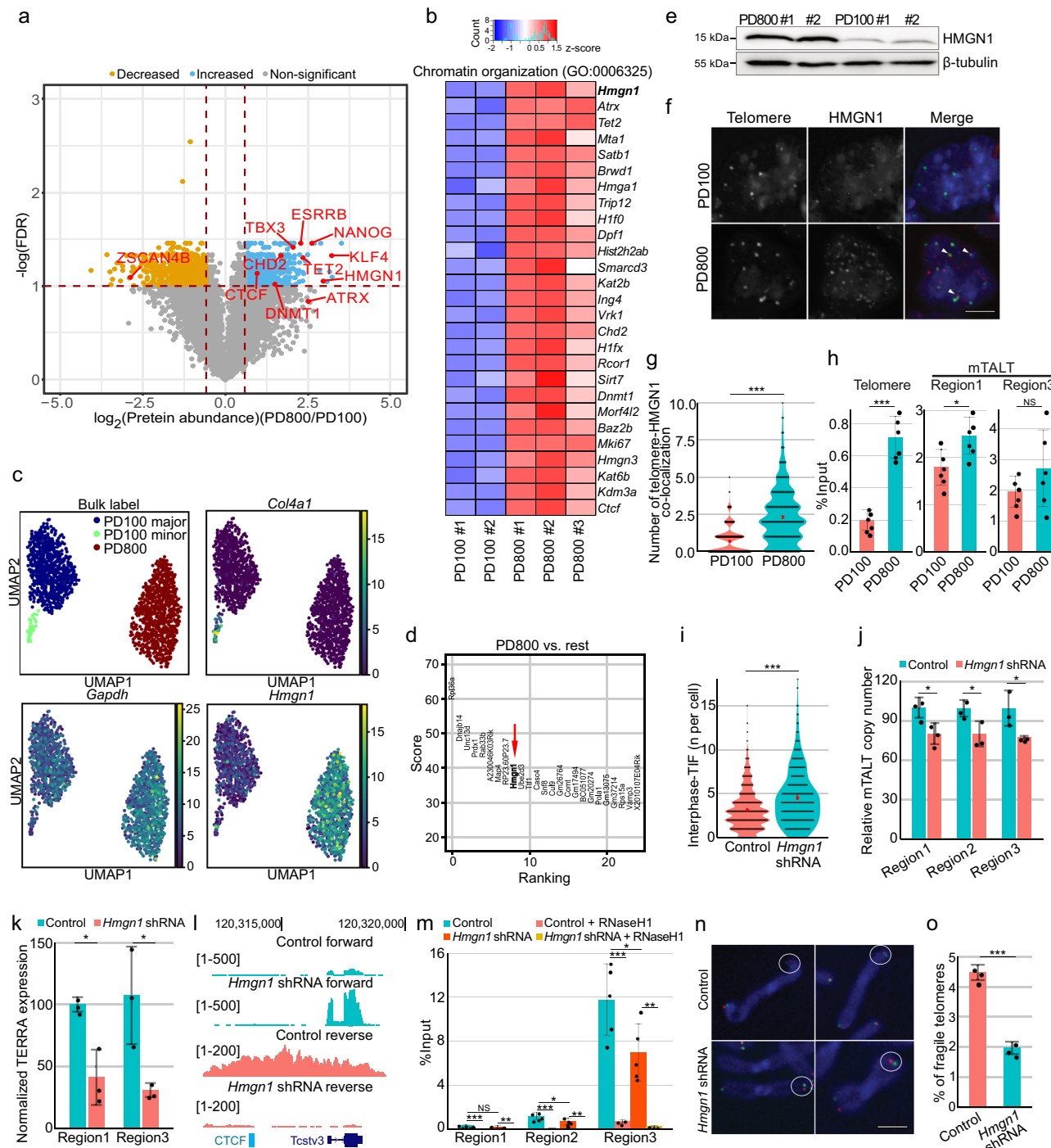

## Discussion

In this study, we established an mESC ALT model in which the non-telomeric unique sequence, mTALT, was utilised for telomere maintenance. By tracking the appearance of ALT cells, we showed that mTALT can be duplicated even before senescence occurs and that mTALT-duplicated cells were selected when overcoming the senescence. The upregulated HMGN1 in post-ALT cells stimulates the increase of TERRA and R-loops possibly through changes in local telomeric chromatin. R-loop-induced telomere instability may lead to telomere lengthening and chromosome maintenance (Fig. 6).

Telomeres are adequately protected under normal conditions, but when damage occurs they can undergo dynamic changes such as external sequence insertions. In various species, specific

sequences have been known to change telomere composition. The presence of mTALT after short telomeric repeats on chromosome 11 in telomerase-proficient strains strongly suggests that mTALT has the potential to repair telomere damage that could not be repaired by telomerase. This is the first description of a specific mammalian ALT template that can protect telomeres even when telomerase activity exists and that can eventually be incorporated into telomeres. Additionally, the fact that cellular senescence did occur before ALT was fully activated suggests that mTALT-driven telomere repair was not fast enough to protect shortened telomeres and prevent senescence.

Among the numerous subtelomeric sequences, mTALT has attributes that favour telomere maintenance hence it may be selected for that function. The subterminal location and telomeric

**Fig. 5 Multi-omic analyses identified HMGN1 as a protein involved in ALT. a** Volcano plot showing the differentially expressed proteins in post-ALT cells (PD800) compared to pre-ALT cells (PD100). Quantitative proteomic analysis was done in biological triplicate for PD800 and duplicate for PD100. The red dot lines indicate 1.5 fold change lines (vertical), and 0.1 of false discovery rate (FDR) line (horizontal). **b** Heat map showing the differentially expressed proteins of the term chromatin organisation. Colour index indicates z-score of protein expressions and 'count' (with regard to histogram) indicates the accumulated number of indicated values represented in the heatmap. **c** UMAP projection of single-cell RNA-seq data. Each dot represents a different cell. Each plot is colour-coded by identified clusters (top left), expression level of *Col4a1* (top right), *Gapdh* (bottom left), and *Hmgn1* (bottom right). **d** Genes were ranked according to their contributions to differentiation of the PD800 cluster from the others. **e** Western blot showing HMGN1 expression levels in post-ALT (PD800) and pre-ALT (PD100) cells. Tubulin was used as a loading control. **f** Representative image of immunostaining of HMGN1 and telomeres. Scale bar, 5 μm. **g** Quantification of HMGN1 and telomere co-localisation. The dots and bars represent means and confidence intervals for the means, respectively, from ≥200 cells per condition over three independent experiments. P value from a two-tailed unpaired t-test: ***P < 0.0001. **h** Chromatin immunoprecipitation (ChIP) analysis with HMGN1 antibody. The bars represent the means and SDs from three biologically independent replicates. P value from a two-tailed unpaired t-test: ***P < 0.0001, *P = 0.0130, NS non-significant. **i** Violin plot showing the TIFs of control and HMGN1 knock-down cells. Each dot represents TIFs in a cell. The dots and bars represent means and confidence intervals for the means, respectively, from ≥200 cells per condition over three independent experiments. P value from a two-tailed unpaired t-test: ***P < 0.0001. **j** Telomere length quantification with mmqPCR. The bars represent the means and SDs from three biologically independent replicates. P values from a two-tailed unpaired t-test: Region 1 *P = 0.0371, Region 2 *P = 0.0433, Region 3 *P = 0.0466. **k** TERRA expression quantified by qPCR. All infected cells were post-ALT (PD800). P values from a two-tailed unpaired t-test: Region 1 *P = 0.0116; Region 3 *P = 0.0293. **l** Snapshot of the mTALT region of RNA-seq. 'forward' and 'reverse' denote the transcription directions. **m** DRIP-qPCR quantified with specific primers. The bars represent the means and SDs from five biologically independent replicates for no RNaseH1-treated samples and three for RNaseH1-treated samples. All infected cells were post-ALT (PD800). P values from a two-tailed unpaired t-test: Region 1 NS, non-significant, ***P = 0.0009, **P = 0.0040; Region 2 *P = 0.0331, ***P = 0.0009, **P = 0.0079; Region 3 *P = 0.0332, ***P = 0.0013, **P = 0.0050. **n** Representative images of telomere fragility. Scale bar, 1 μm. **o** Telomere fragility was quantified from chromosome orientation (CO) FISH. The bars represent the means and SDs from ≥30 cells per condition over three independent experiments. P value from a two-tailed unpaired t-test: ***P = 0.0002. Regions 1–3 denote specific regions inside the mTALT sequence. Source data are provided as a Source Data file.

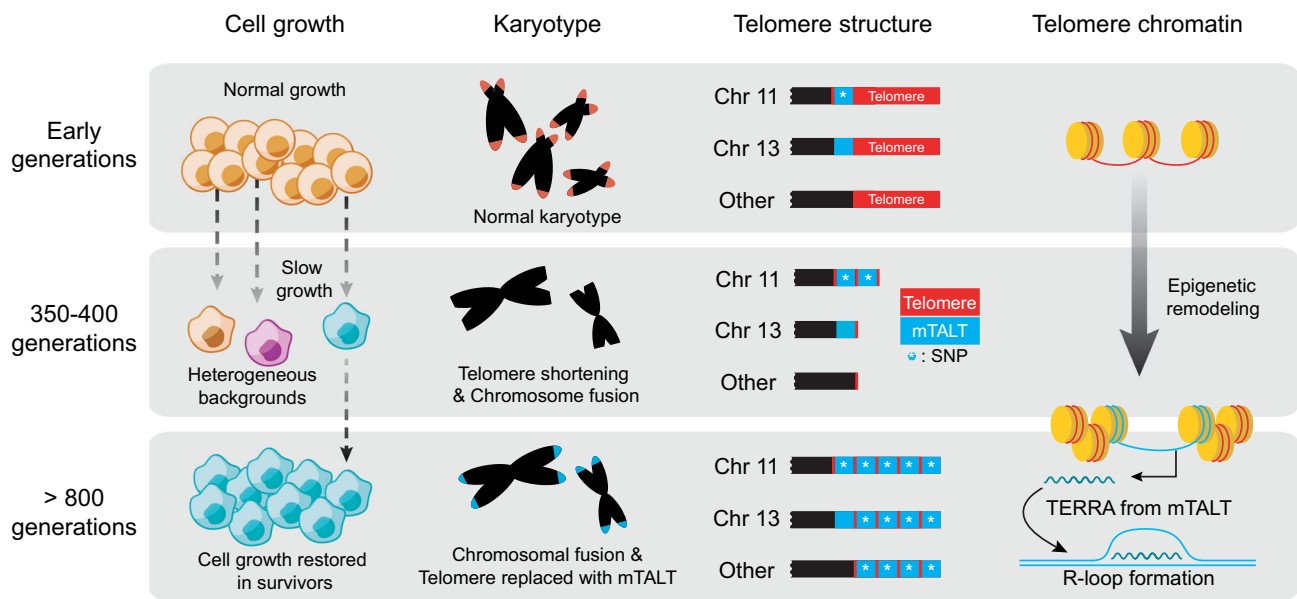

**Fig. 6 A working model of ALT activation of mESCs.** This figure summarises the emergence process of ALT mESC from various aspects. After the Terc gene was knocked out, cells of the early generation grew normally, but as the telomere length decreased growth slowed down and cells approached a senescent state. Affected by telomere shortening, genome-wide copy-number changes, and chromosomal fusions occurred, and transcriptional networks and cellular physiology changed accordingly. One or more cells activated ALT and were selected to form a homogenous post-ALT cell population. In an evolutionary time-line, the mTALT sequence was located at chromosome 13 first, and at a certain point, replicated at the end of chromosome 11. In the process of telomere shortening, the only mTALT of chromosome 11 underwent selective duplication, and during ALT activation, the mTALT of chromosome 11 was amplified in *cis* and *trans* to cover all telomeres. Through transcriptome and proteome analyses, we confirmed that genome-wide epigenetic remodelling occurred during the ALT activation process, and the expression of telomeric transcripts and R-loop formation increased due to HMGN1-dependent telomeric changes, contributing to telomere maintenance.

repeat-flanked character of mTALT may aid its recruitment, which may result in the preferential use of the mTALT on chromosome 11 over that on chromosome 13 for building new telomeres. In humans, tags flanked by telomeric repeats are reported to have the potential to be duplicated to other chromosomes[30]. For example, the SV40 sequence can be integrated into telomeres during immortalisation of the Werner syndrome cell line[14,15]. It is notable that SV40-based ALT cells also did not

show the common ALT marker, APB, which suggests a similar mechanism of ALT to that of ALT mESCs we report here. One difference between the previously reported cases and our findings is that the mTALT sequence naturally exists in the mouse genome as a source of templates for ALT telomeres, which underscores its cell-intrinsic potential of reshaping telomeres. Furthermore, TALT elements in *C. elegans* and Y′ elements in *S. cerevisiae* are also located in subtelomeric regions. Chromatin structure also

seems to be a determinant for selection as a template for telomeres. Various templates used in *S. pombe* HAATI survivors have a heterochromatic nature. The amplified mTALT also contains an overall heterochromatin structure except for a CTCF-binding site. Non-canonical telomere examples in other organisms show similarities with mTALTs. This suggests that the telomere lengthening mechanism described in this study may be universal.

During telomere shortening, LOH was strongest at the subtelomeric region of chromosome 11. This implies that there is a tendency for homologous recombination to occur between homologous chromosomes rather than between sister chromatids. There may have been a shift towards non-allelic homologous recombination to create a favourable environment for mTALT amplification. During these hypothetical rearrangements, mTALT was likely *cis*-duplicated before ALT activation. Furthermore, CNV occurred on chromosome 11 while other chromosomes were not affected. It should be interesting to address why a specific chromosome went through such intense rearrangements during telomere shortening and how these changes affect the ALT activation process.

The analysis of CNVs using bulk WGS data showed that highly heterogeneous pre-ALT PD350 cells underwent selection to become homogeneous post-ALT PD450 cells (Fig. 3d). At PD350, tens of millions of cells were under telomeric crisis with genomic aberrations, and a certain proportion of the cells would share ALT potentials. The selection upon ALT activation was likely to be multi-cellular events even though the events would not simultaneously happen. However, the genomic structure at PD450 and PD800 also can be shaped from monoclonal ALT activation followed by the clonal expansion, so bulk WGS data have limitations in revealing the clonal evolution at the level of single cells after PD350 upon ALT activation. Therefore, further studies using longitudinal single-cell WGS data will be necessary to trace the monoclonal or polyclonal evolution and lineage of ALT-activated cells.

It is noteworthy that there is a natural variation in the copy number of mTALT. During telomere shortening, genomic heterogeneity was increased in terms of copy numbers. mTALT may have gone through similar copy-number heterogeneity around senescence and cells containing several mTALTs may have a higher chance of being selected in ALT activation. The naturally increased copy number of mTALT in some strains may be a sign of repairing telomere damage which could not be handled by telomerase. In the Hawaiian strain of *C. elegans*, the duplication of TALT at the telomeric end was a prerequisite for the introduction of TALT-mediated ALT[13]. Our finding of ALT mESC may have been possible due to starting with a specific strain bearing the right template at the right place.

It is conceivable that telomere instability remains after ALT is activated given the continuing DNA damage in telomeres and an increase in CNV. Remarkably, CNVs occurred mainly in the long arm of chromosomes, where telomere-shortening-dependent chromosomal fusions rarely happen. Although CNVs in chromosome fusion junctions have been attributed to the DNA replication-based mechanism called fork stalling and template switching, the exact mechanism by which CNVs occurred with telomere lengthening remains unresolved. The resulting stochastic CNV changes can be passengers or drivers depending on whether or not they provide better fitness. The copy number increase in *Hmgn1* is a good example, located in the duplicated subtelomeric region. Considering that the expression of *Hmgn1* is uniformly increased in most PD800 cells, we can hypothesise that cells with high HMGN1 levels were selected to constitute a significant proportion of the ALT cell population. After the initiation of ALT, the genome may continuously change until the telomeres are fully stabilised. As a result, telomere instability may

promote not only telomere synthesis but chromosomal rearrangements.

Interestingly, the mTALT sequence shows a high level of transcriptions despite the heterochromatic nature of telomeres. ALT telomeres have been thought to promote telomeric recombination and transcription by loosening the heterochromatin structure of telomeres[31]. However, a recent report revealed that the heterochromatic nature of telomeres can induce transcription of TERRA and the appearance of ALT characteristics[32]. We showed that the mTALT-based telomeres also appeared as heterochromatic, and at the same time, produced a number of transcripts. These findings suggest that mTALT may have characteristics that favour transcription, which may affect the template-selection process during ALT activation.

Increased HMGN1 and TERRA contribute to ALT maintenance. Specifically, HMGN1-dependent reverse transcripts of new telomeres are produced and they subsequently form R-loops. R-loops can be generated during abrupt changes in transcriptional demands or when transcription is blocked[22]. The formation of telomeric R-loops of ALT mESC can also be explained by the chromatin decompaction effect produced by HMGN1 which increases telomere transcription, and the loosened chromatin structure which may assist the interaction of transcribed RNA and DNA. In addition, the increased telomere damage level after ALT activation may interfere with transcription progress and promote the generation of co-transcriptional R-loops. Reasons for the increased telomere damage after ALT activation could not be accurately determined but it may be due to the lowered shelterin density which promotes stable telomere replication and protection. Another interesting aspect is that the formation of R-loops can be increased when there is a conflict between DNA replication and transcription machinery (so-called transcription–replication collisions, TRCs)[33]. While telomere replication proceeds from the subtelomere to the end of the chromosome, the direction of mTALT transcription is the reverse. Head-on TRCs specifically promote R-loop formation. Thus, the association between the mTALT sequence characteristics and HMGN1 function produced telomeric R-loops and regulated telomere physiology.

R-loops can induce genome instability by interfering with transcription and replication. In particular, DSB formation and DNA loss may occur if a fork collapse occurs following replication fork stalling. However, several reports allude to potential R-loop contribution to the maintenance of the genome, particularly the telomere. This is possible by chromatin regulation[34], priming DNA replication[35] and promotion of inter-telomere HR[36]. Of note, R-loops can prevent telomeric replication fork collapse through HR which prevents telomeres from becoming dysfunctional[37]. In a study of ALT cancer cells, when the amount of RNaseH1 was depleted or overexpressed, abrupt telomere shortening occurred[38]. In other words, the precisely controlled telomeric R-loop makes an important contribution to telomere stability without seriously harming telomeric integrity.

Although there have been reports that the ALT-like mechanism is involved in telomere maintenance during mouse development, the mouse organismal model that maintains telomeres only by ALT has not been established. Considering that telomeres of mice are considerably longer than those of humans, it is still unclear whether the mechanism of telomere maintenance found in mice can be applied universally to various mammals including humans. In addition, further studies are needed to determine whether the ALT phenomenon identified in this study works in the same way as in human ALT cancer cells in which mechanisms based on homologous recombination or break-induced replication operate.

Our report is the first in mammalian models to longitudinally analyse telomeres maintained with non-telomeric sequences.

Considering that this phenomenon is widely conserved in various model organisms such as *S. cerevisiae*, *S. pombe*, *C. elegans* and the specific human immortalised cell line (AG11395), the characteristics of ALT revealed in this study will contribute to the expansion of our knowledge on telomere biology.

## Methods

**Cells**. Terc knock-out mESCs were kind gifts from H. Niida. The generation and culture method was described previously[39]. Briefly, the Terc targeting construct (TR-10) was transfected into E14 mESC by electroporation to exchange the first Terc allele. The targeting plasmid included PGK-neo gene flanked by loxP sites. The transfected cells were selected with G418 (200 mg/ml) and Ganciclovir (1 mM). The resistant Terc +/− cells were picked after 8 days. The neo gene was removed by infection with the adenovirus of Cre recombinase. The Terc +/− cells were re-targeted with TR-10. Two lines of Terc−/− ESC, DKO301 and DKO741 were recovered. In the next paper, the authors found ALT-activated mESC after ~450 passagings. Both of DKO301 and DKO741 produced survivors with rare frequencies[16]. We got those ALT survivor cells at each PDL timing (PD100, PD350, PD400, PD450) with control (Terc+/+) cells. For mESC maintenance and stock preparation, mouse embryonic fibroblast (MEF) cells were used as feeder cells. Feeder-free mESCs were used for all experiments including nucleic acids extraction, protein extraction, immunostaining, and fluorescent in situ hybridisation.

**Cell growth assay**. The same number of cells ($1.5*10^4$ cells/well) were seeded in a 48-well plate in triplicate. Four plates were seeded simultaneously for counting of each day. Six hours after seeding, the plate was washed with 1× PBS. Each well is fixed with 4% paraformaldehyde in PBS and washed with water. Fixed cells were stained with 0.1% crystal violet in 10% ethanol solution. After wash with water and air-dry, each plate was kept until imaging. The staining was repeated at daily intervals, and cell numbers were calculated on the images with Image J (version 1.53) software.

**TRF assay**. TRF assay was done based on previously reported way[13]. Briefly, 5 μg of genomic DNA of each time point was cut with *AluI*/*MboI* or *HinfI*. Each DNA was run on 0.7% agarose gel at 25 V for 8 h. After depurination with 0.25 M HCl, denaturation and neutralisation, DNA was transferred to nitrocellulose membrane. Telomeric signal was visualised with DIG luminescent detection kit (Roche) according to manufacturer's guides.

**C-circle assay**. C-circle assay was done based on previously reported way with a few modifications[40]. We did not use restriction enzymes and Exonuclease to preserve a potential mTALT-based C-circles. Two kinds of probes were used to detect C-circle signals. One was the telomeric probe dig-(CCCTAA)*4, and the other was the mixture of mTALT-specific probes which were established with DIG DNA labelling mix (Roche) according to manufacturer's guides. Primers used for making mTALT-specific probes were listed in Supplementary Table 1. Φ29 polymerase-amplified reactants were blotted onto nitrocellulose membrane with slot-blotter and visualised with DIG luminescent detection kit (Roche) according to manufacturer's guides. Images were acquired with Image Quant LAS 4000 mini.

**Analysis of whole genome sequencing**. Genomic DNA was extracted using DNeasy Blood & Tissue Kit (QIAGEN). Macrogen (South Korea) performed library preparation and sequencing using NovaSeq 6000 (Illumina) platform. 151 bp paired-end reads were produced for all the samples. The reads were mapped to the mm10 reference genome using Burrows–Wheeler Aligner (BWA) MEM algorithm (version 0.7.17)[41]. Duplicated reads were marked with Picard-Tools (version 1.96) and Samtools (version 1.9)[42]. For copy number calculation, two matched BAM files were compared with a window size of 1 kb using ngCGH (version 1)[43]. Then, the copy number altered regions were defined by segmentation of the genome using DNA-copy (version 1.52)[44]. The circus plot for the CNVs was drawn by BioCircos (version 0.3.4)[45]. To visualise mTALT reads on the UCSC genome browser[46], chr13: 119,891,001–120,300,000 region on the mm10 reference genome was masked with Ns due to partial sequence duplicates within mTALT to the region, then all the reads were re-mapped onto the masked genome. SNPs and insertion and deletions (INDELs) were detected and genotyped according to the assembled E14 genome using Haplotyper Caller (GATK version 3.8-0; filtering option = QD < 2.0||FS > 60.0||MQ < 40.0||MQRankSum < −12.5||ReadPosRankSum < −8.0)[47,48]. The SNPs and INDELs were annotated using ANNOVAR (2018Apr16)[49], and the functional enrichment of the genes was examined using DAVID (version 6.8)[50]. Paired-end reads including TTAGGG or CCCTAA at least three times, respectively, in either read were extracted from the raw-sequencing data for the analysis of telomere-flanking structure and directionality of mTALT.

**Analysis of ATAC-seq**. A basic process is from a previous paper[20]. 50,000–75,000 cells were pelleted at 500×*g*, 4 °C for 5 min. Cells were treated with 50 μl of cold RSB (10 mM Tris–Cl, pH 7.4, 10 mM NaCl, 3 mM MgCl₂) containing 0.1% NP40, 0.1% Tween-20, and 0.01% Digitonin and pipetted three times. The cells were

incubated on ice for 3 min, and washed with 1 ml of cold RSB containing 0.1% Tween-20 but no other detergents. The cells were pelleted and resuspended with 50 μl of transposition mixture (25 μl 2× TD buffer, 2.5 μl transposase, 16.5 μl PBS, 0.5 μl 1% digitonin, 0.5 μl 10% Tween-20, 5 μl H20) (2× TD buffer = 20 mM Tris–Cl, pH 7.6, 10 mM MgCl₂, 20% dimethyl formamide). The reaction mixture was incubated at 37 °C for 30 min with consistent shaking, and stopped by adding 50 μl of tagmentation stop buffer (10 mM Tris–Cl, pH 8.0, 20 mM EDTA, pH 8.0). The reaction was cleaned with Zymo DNA clean and concentrator-5 kits. DNAs were eluted in 21 μl of elution buffer, and amplified for five cycles using NEBnext high-fidelity 2× PCR master mix. Using 5 μl (10%) of the pre-amplified mixture, a 15 μl qPCR was performed to determine the number of additional cycles. The linear relative fluorescence versus cycle was plotted, and the cycle number that fitted to 1/4 of maximum fluorescence intensity was determined. The remaining 45 μl of PCR reaction was processed with qPCR with additional cycles determined. The final PCR reaction was purified using Zymo kits. Sequencing was performed with NovaSeq 6000 platform by 101 bp paired-end. The reads were pre-processed in the same way as the reads of WGS until production of sorted BAM files. Peak-calling was performed using MACS2 (version 2.1.2) callpeak command with parameters "–shift −100 –extsize 200 –nomodel"[51]. All the significant peaks ($0.05 \leq q\text{-}value$) across the samples were merged with BEDTools (version 2.25.0)[52], then filtered by the ENCODE mm10 blacklist (https://www.encodeproject.org/files/ENCFF547MET). According to the significant peak regions, ATAC-seq reads were counted, and the count table was used to identify differential peaks using R/Bioconductor DESeq2 (version 1.18.1) with a cut-off of q-value = 0.01 and fold change 2 in R (version 3.4.3)[53]. The read counts were log2-transformed, and unsupervised hierarchical clustering was performed with the top 10% peaks ranked by variance. Chromatin states of E14 cells were characterised by chromHMM[54,55], and average profiles of ATAC-seq peaks per chromatin state were drawn using deepTools (version 3.3.0)[56].

**Analysis of long and short read bulk RNA sequencing**. mESCs were harvested in Trizol. RNA was extracted using chloroform and isopropanol precipitation. Macrogen (South Korea) performed library preparation and sequencing process using HiSeq 4000 (Illumina). Biological duplicate samples were sequenced by 101 bp paired-end. The reads were aligned to the mm10 reference genome using the STAR aligner (version 2.5.3a)[57]. Fragments per kilobase million (FPKM) and transcripts per kilobase million (TPM) were calculated with RSEM (version 1.3.0)[58]. Differentially expressed genes (DEG) were identified by DESeq2 (version 1.18.1). Using the same RNA sequenced by short-read technology, long-read Iso-Seq library was constructed using the SMATer PCR cDNA Synthesis kit and DNA Template prep kit 1.0. The library was sequenced with the PacBio Sequel System then analysed using SMRT Link v6.0.

**Single cell sequencing and analysis**. Cells were encapsulated with a custom inDrops microfluidic setup as previously reported with the following modifications[59]. In RT-lysis mix, Maxima H-RTase was used instead of Superscript III and additional oligos (RP-TSO, IDT) were added at concentration of 3 μM to facilitate template switching reaction within a droplet. After exoI treatment, cDNA was recovered using 0.8× AMPure beads and amplified with P7 and PE1 primers using KAPA HiFi Hotstart ReadyMix (KAPA Biosystems). The PCR programme was 98 °C for 3 min, 14 cycles of [98 °C for 20 s, 63 °C for 30 s, 72 °C for 3 min], 72 °C for 10 min, and the product were purified with 0.8× AMPure beads. Then, tagmentation reaction was performed with Nextra XT DNA library kit following the Drop-seq protocol[60] and the library was generated by amplifying the purified product with P7 and indexed P5 primers using KAPA HiFi Hotstart ReadyMix. The library was quantified using NEB Library Quant Kit (NEB), and sequenced with 10% PhiX spike-in on a Illumina HiSeq 2500 using a rapid run mode with the following specifications: 50 cycles for Read1, 8 cycles for Index1, 50 cycles for Read2, 8 cycles for Index2.

The raw files were processed to count matrix using the inDrops pipeline (https://github.com/indrops/indrops). Briefly, library index was identified with Index2, cell barcode was inferred from Index1 and Read2, and transcriptome counts were quantified by mapping to the reference transcriptome built from the mouse GRcm38/mm10 genome assembly file. Other settings were set default. The resulting count matrix was then analysed using the Scanpy package (version 1.6.0) (https://github.com/theislab/scanpy)[61]. 1664 cells (799 in PD100 and 865 in PD800) were identified after removing cells that has a total reads fewer than 1000 or >20% mitochondrial content. Variable genes were selected using the gene expression variation model[62] with a threshold of FDR < 0.05, which illustrates relationship between mean and coefficient variation (CV). The Poisson distribution of counts in individual gene was estimated by its sample mean. For clustering, the top 80 principal components were used to construct the shared nearest-neighbour graph using the parameter 'n_neighbors' of 15 followed by the Louvain clustering and UMAP visualisation in the Scanpy package. Primers used were listed in Supplementary Table 1.

**Telomere counting analysis**. Telomere length was quantified from FASTQ files using Computel (version 1.2) with default parameters[63]. Accessibility of telomeric repeats was quantified from ATAC-seq FASTQ file using Computel (version 1.2) with default parameters. Variant telomere content was determined by

TelomereHunter (version 1.1.0) with default parameters[64]. Pair and Single telomere-containing reads was counted by Telomerecat (version 3.2) with default parameters[65].

**mTALT copy number estimation**. Genomes of 16 laboratory strains of mouse were downloaded from Mouse Genomes Project (https://www.sanger.ac.uk/science/data/mouse-genomes-project). mTALT copy number was estimated by calculating the normalised coverage of mTALT regions (Chr. 13:120311821–120319190). Normalised coverage was calculated by dividing the depth of coverage within mTALT regions by the depth of coverage of the single copy region in same chromosome block (Chr. 13:109998778–110006148).

**Monochrome multiplex quantitative PCR (mmqPCR)**. mmqPCR was modified from a previously published paper[66]. mmqPCR was used for telomere and mTALT length measurement. This technique can measure Ct (cycle threshold) value for two amplicons in a reaction well with a single DNA-intercalating dye. The Ct for more abundant target can be measured at earlier cycles, and at later cycles the Ct for the second target can be collected at a temperature highly above the melting temperature of the first amplicon. Primers for second amplicon were placed with GC-clamps on both ends to increase the melting temperature. Reaction mixture was 1× HOT FIREpol EvaGreen qPCR Supermix (Solis biodyne), target primers (final concentrations 500 nM each), single copy gene control primers (final concentrations 500 nM each), template DNA (100 ng). The thermal cycling programme for telomere was Stage 1: 15 min at 95 °C; Stage 2: 2 cycles of 15 s at 94 °C, 15 s at 49 °C; and Stage 3: 33 cycles of 15 s at 94 °C, 10 s at 62 °C, 15 s at 74 °C with signal capture, 10 s at 84 °C, 15 s at 88 °C with signal capture. The thermal cycling programme for mTALT was Stage 1: 15 min at 95 °C; Stage 2: 35 cycles of 15 s at 94 °C, 20 s at 62 °C, 15 s at 72 °C with signal capture, 10 s at 84 °C, 15 s at 88 °C with signal capture. Primers used were listed in Supplementary Table 1.

**Interphase TIF assay**. The protocol for Immunofluorescence and telomere fluorescence in situ hybridisation (FISH) was modified from a previous paper[67]. The cells were cultured in feeder-free state and trypsinized at confluency of 50–60%. The cells were diluted with PBS at concentration of $2–5 \times 10^5$ cells/ml. The diluted cells were cytocentrifuged with the appropriately assembled cytospin funnels at $200 \times g$ for 10 min. The cells on slide were fixed with 3.7% formaldehyde, and permeabilized with KCM buffer (120 mM KCl, 20 mM NaCl, 10 mM Tris–Cl, pH 7.5, 0.1% Triton X-100). ABDIL buffer containing RNaseA (100 ug/ml) were added and incubated for blocking. The primary antibody (γ-H2A.X, Biolegend #613401, 1/1000 dilution) diluted in ABDIL (20 mM Tris–Cl, pH 7.5, 2% bovine serum albumine, 0.2% fish gelatin, 150 mM NaCl, 0.1% Triton X-100, 0.1% sodium azide) was added and incubated overnight at 4 °C. The slides were washed with PBST and incubated with ABDIL-diluted secondary antibody for 1 h at room temperature. The slides were washed with PBST again. The cells were fixed with 3.7% formaldehyde. The slides were ethanol dehydrated with a graded ethanol series: 70%, 90%, and 100% ethanol. After the slides were dried, telomere-PNA probe diluted in hybridisation solution (70% formamide, 0.5% blocking reagent, 10 mM Tris–Cl, pH 7.5) at 0.3 μg/ml was added and the cells were denatured for 5 min at 80 °C. The hybridisation was done overnight at room temperature. The slides were washed with PNA wash A (70% formamide, pH 7.5, 10 mM Tris–Cl, pH 7.5) and B (50 mM Tris–Cl, pH 7.5, 150 mM NaCl, 0.8% Tween 20) buffer. After ethanol dehydration and air-dry, the slides were mounted with Prolong Gold and store overnight before imaging. In HMGN1 staining, the overall process was same but HMGN1-specific primary antibody (a kind gift from Michael Bustin) was used. Images were acquired with softWoRx (version 6.11). Co-localisations were measured with MatCol (version 0.95).

**Metaphase-immunostaining**. Basically, metaphase-immunostaining was same as interphase assay except chromosome preparation step. The cultured cells were treated with 1 μM nocodazole for 1.5 h before harvest at 50–60% confluency. The cells were trypsinized and washed with PBS. The cells were resuspended with 0.8% sodium citrate solution and incubated at 37 °C for 5 min. The swelled cells were cytocentrifuged at $400 \times g$ for 10 min. The remaining steps were same as interphase-immunostaining. TRF2 antibody (Novus Biologicals #NB110-57130) was diluted as 1/1000.

**Quantitative-FISH (Q-FISH)**. The overall process was performed as previously reported[67]. The cultured cells were treated with 1 μM nocodazole for 1.5 h before harvest at 50–60% confluency. The trypsinized cells were resuspended with 0.8% sodium citrate solution and incubated at 37 °C for 10 min. After spin-down, the supernatant was aspirated and the cells were resuspended in ~300 μl of remaining solution. 1 ml of pre-cooled fixative (75% methanol and 25% acetic acid) was added dropwise and the tubes were tapped after each drop. 9 ml of fixative was added with consistent tapping. The cells were fixed at 4 °C overnight and stored at −20 °C. Slides were prepared by soaking in 100% ethanol for 1 h. The cleaned slides were air-dried and the fresh fixative was prepared and cooled. The fixed cells were spin-down and resuspended in fresh fixative at $~1 \times 10^6$ cells/ml. Five paper towels were soaked with water and the cleaned slides were placed onto the wet towels. The resuspended cells were dropped from a height of 2 cm onto the slides. The slides were incubated

at 65 °C for 1 min, and allowed to air-dry. The slides were checked with bright-filed light microscope to look for good metaphase chromosomes. The slides were cured overnight at room temperature. The slides were fixed with 3.7% formaldehyde and incubated with PBS containing RNase A (250 μg/ml). The slides were re-fixed with 3.7% formaldehyde and washed with PBS. The slides were ethanol dehydrated with a graded ethanol series: 70%, 90%, and 100% ethanol. After the slides were dried, telomere-PNA probe diluted in hybridisation solution at 0.3 μg/ml was added and the cells were denatured for 10 min at 70 °C. The remaining washing and mounting steps were same as the immunostaining protocols.

**Chromosome orientation-FISH (CO-FISH)**. The overall process was performed as previously reported[67]. Cells were seeded and incubated at 40–50% confluency. The cells were incubated for 12 h in growth media with 10 μM 3:1 BrdU/BrdC. For the last 1.5 h with BrdU/BrdC, 1 μM nocodazole was added to media. After spin-down, the supernatant was aspirated and the cells were resuspended in ~300 μl of remaining solution. The process of cell harvest, fixation, and dropping onto slides was same as Q-FISH. The cured slides were rehydrated with PBS for 5 min and fixed with 3.7% formaldehyde for 5 min at room temperature. After PBS wash, the slides were incubated with PBS containing RNase A (250 μg/ml) for 15 min at 37 °C. The slides were washed with PBS and rinsed with 2× SSC, which is followed by incubation in 2× SSC containing 0.5 μg/ml Hoechst 33258 for 15 min in the dark. The slides were irradiated with ~365 nm UV light for 30 min and washed with distilled, deionized water. 80 μl of 10 U/μl of ExonucleaseIII were treated to the slides and incubated for 30 min at 37 °C. During this step, the newly synthesised DNA strands were degraded. After wash with PBS, 70% deionized formamide/30% 2× SSC solution was added and the slides were incubated on a heating block at 70 °C for 10 min. After rinse with water, the slides were ethanol-dehydrated. TelG-Cy3 PNA probe was placed onto the slides and denatured at 70 °C for 10 min, and then hybridisation was done for 2 h at room temperature in the dark. The slides were washed twice in PNA wash A buffer, and TelC-FITC PNA probe was loaded. The slides were hybridised for 2 h at room temperature in the dark, and washed twice in PNA wash A buffer, and three times in PNA wash B buffer. The slides were rinsed with water and air-dried completely. The samples were mounted with Prolong Gold (Invitrogen #P36930). Images were acquired with softWoRx (version 6.11).

**Lentivirus-based gene knock-down**. HEK293FT cells (Takara #632180) were plated at 100 mm plates. HEK293FT cells were co-transfected with target shRNA-expressing transfer vector (PLKO.1 backbone), pMD2.G (Addgene #12259), and psPAX2 (Addgene #12260) by PEI method. In 100 mm plate, the actual ratio of each vector and PEI is that transfer vector: pMD2.G:psPAX2:PEI = 6:3:1.5:21 (μg). After 7–12 h, media was changed with fresh one. After 36–48 h, culture media was harvested and filtered with 0.45 μm PVDF syringe filter. Filtered media was incubated with Lenti-X concentrator (Clontech #631232) at 5:1 ratio at 4 °C on rotator for 1–2 days. The virus soup with concentrator was centrifuged at $1500 \times g$, 45 min. The resulted virus pellet was resuspended with ~200 μl fresh culture media. The overall virus harvest process was repeated after 24 h from first harvest. ALT mESC was plated at 12-well plate. At ~50% confluency the resuspended virus pellet was added to target mESC with poly-b-rene (Sigma-Aldrich #107689) at concentration of 4 μg/ml. After 24 h second harvested virus was infected in the same way. After 24 h of incubation the infected cells were transferred to new plate. When the transferred cells reached 80–90% confluency, puromycin (gibco #A11138-03) was added to each well at 1 μg/ml. The selecting media was changed every day. The decreased expressions of target genes were confirmed by qPCR or western blot. Primers used for cloning transfer vectors were listed in Supplementary Table 1.

**TERRA knockdown**. TERRA knockdown with LNA gapmer (Qiagen, custom ordered) was performed as previously reported[68]. mESCs were grown until 50–60% confluency at six-well plates in triplicates. mESCs were transfected with LNA gapmers (TERRA and control) at a final concentration of 4 nM with Lipofectamine 3000 reagent (Thermo Fisher #L3000001) following manufacturer's guide. After 24 h, cells were harvested and analysed.

**Sample preparation for quantitative proteomics analysis**. $~1*10^7$ cells were harvested at 70–80% confluency from 100 mm plates at two PDL time points (PD100 in duplicate and PD800 in triplicate). The pelleted cells were incubated with hypotonic buffer A (10 mM HEPES, pH 7.5, 20 mM NaCl, 0.01% Triton X-100, 1 mM DTT, 1 mM PMSF, and proteinase inhibitor cocktail) and pipetted up-and-down ~100 times. The nuclei were pelleted and washed once with PBS. The nuclei samples were then resuspended with 8 M urea in 50 mM ammonium bicarbonate (ABC) buffer (pH 8.5), and sonicated at maximum intensity for 15 min (Cosmobio bioruptor), and followed by BCA protein assay after centrifuging at $14,000 \times g$ for 15 min at 4 °C. The nuclear lysates were reduced by 10 mM DTT for 1 h at 37 °C and followed by alkylation with 40 mM iodoacetamide for 30 min at 37 °C in the dark. Samples were diluted with 50 mM ABC to be <1 M of urea concentration and followed by trypsin (Thermo Fisher, MS grade) digestion with 2% (w/w) at 37 °C for overnight. Remaining activity of protease was quenched by acidifying the sample with 0.4% TFA and the resulting digests were desalted by C18 SPE cartridge (Supelco).

Equal amount of digests for each sample, reconstituted with 50 mM HEPES (pH 7.5) buffer, was then labelled with TMT-10plex reagents according to the manufacturer's protocol as follows: TMT-126 and 127N for DKO741 cells at PD100 in duplicate; TMT-127C, 128N, and 129N for DKO741 cells at PD800 in triplicate. All TMT-labelled samples were combined into a single sample, and followed by mid-pH (pH 8) RPLC off-line fractionation using a 1290 UHPLC (Agilent). The eluent was automatically collected into 96-well plate per each minute and then concatenated to 24 fractions.

**Liquid chromatography and tandem mass spectrometry.** The TMT-labelled 24 fractions were analysed using an Orbitrap Fusion Lumos mass spectrometer (Thermo Fisher Scientific) coupled with a nanoACQUITY UPLC equipped with an in-house packed trap (150 μm i.d. × 3 cm) and analytical column (75 μm i.d. × 100 cm) using 3 μm of Jupiter C18 particle (Phenomenex). A linear gradient of solvent A (water with 0.1% formic acid) and solvent B (ACN with 0.1% formic acid) in ACN) was applied at a flow rate 300 nL/min: 5–10% solvent B for initial 5 min and 10–40% solvent B for next 175 min. Full MS scans ($m/z$ 350–1500) were acquired at a resolution of 120k (at $m/z$ 200) with 5E5 of AGC target value and 100 ms of ITmax. Selected precursor ions were first isolated at 0.7 Th of isolation window and subjected to HCD fragmentation for MS2 scans in orbitrap at a resolution of 15k (ITmax 60 ms, AGC 1E5 and NCE 30%). The 10 most intense MS2 fragment ions were synchronously isolated in ion trap for final HCD MS3 scans at a resolution of 50k and $2m/z$ of isolation width (AGC 1.2E5, ITmax 150 ms, and NCE 55%). Overall 3 s of cycle time was applied.

**Protein identification and quantitative proteomics analysis.** Proteome Discoverer platform, version 2.1 (Thermo Fisher Scientific), was used for the protein identification and quantitative analysis. Assignment of MS2 spectra was carried out using the SEQUEST algorithm against UniProt mouse proteome (81,557 entries, released 2017-04-25) and the resulting peptide hits were filtered at maximum 1% false discovery rate by Percolator algorithm. Carbamidomethylation of cysteine and TMT-labelling of lysine and peptide N-terminus were set as fixed modifications, while the methionine oxidation was considered as a variable modification. Trypsin specificity with up to two missed cleavage sites was applied. Mass tolerances for precursor and fragment ions were set to 10 ppm and 0.6 Da, respectively. The intensities of TMT reporter ions were adjusted by applying the isotopic correction factors of TMT kit provided by the manufacturer. Only all reporter ions-containing spectra were designated as 'quantifiable spectra'. In order to identify differentially expressed proteins, the signal intensities of all the samples were normalised by quantile normalisation method. After normalisation, Welch $t$-test was performed to compare PD100 and PD800, then the $p$-values were corrected by Benjamini and Hochberg method. For gene set enrichment analysis (GSEA) (version 2.0), genes were ranked by signal-to-noise between PD100 and PD800, and C5 gene sets in MSigDB (molecular signature database) v7 were used for the enrichment score[69].

**DNA–RNA hybrid immunoprecipitation and quantitative PCR.** DRIP protocol was modified from ref. [70]. Nucleic acid was extracted with phenol/choloroform in phase lock gel tubes followed by ethanol precipitation. The purified nucleic acid (DRNA, DNA and RNA) was sonicated at maximum intensity for 15 min (Cosmobio bioruptor). For RNase H-treated control, 10 μg of DRNA was treated with RNase H overnight at 37 °C, and then purified with phenol/chloroform and ethanol precipitation. 10 μg of DRNA was incubated with 5 μg of S9.6 R-loop-specific antibody (Kerafast #ENH001) in 1× IP buffer (10 mM NaPO₄, pH 7, 140 mM NaCl, 0.05% Triton X-100) overnight at 4 °C. 50 μl of Dynabead (ThermoFisher) for each sample was equilibrated with IP buffer and added to IP tubes and incubated at 4 °C for 2 h. The beads were washed three times with IP buffer, and elution buffer (50 mM Tric–Cl, pH 8, 10 mM EDTA, 0.5% SDS, proteinase K) was added. The elution was performed for 3 h at 65 °C. DNA was purified lastly with phenol/chloroform and ethanol precipitation. qPCR was done with the sonicated input DNAs.

**Statistics and reproducibility.** Statistical analyses were performed with R (version 3.6.3) and the results were visualised with R ggplot2 (version 3.3.0) and R bbplot (version 0.2), unless stated otherwise. Statistical significance was assessed by unpaired two-tailed $t$-test unless stated otherwise. Data represent mean ± standard deviation (SD) as indicated in figure legends. $P > 0.05$ was considered not significant. For all experiments, similar results were obtained from at least three biologically independent experiments unless stated otherwise in each figure legend.

**Reporting summary.** Further information on research design is available in the Nature Research Reporting Summary linked to this article.

## Data availability

The ATAC-seq, RNA-seq, whole genome sequencing, and Iso-Seq data are deposited at NCBI GEO: GSE147916. Single-cell RNA-seq data are deposited at NCBI GEO: GSE160487. The mass spectrometry raw data and searching files have been deposited to the ProteomeXchange Consortium with the dataset identifier PXD020419 via MassIVE database (FTP download link: ftp://massive.ucsd.edu/MSV000085772/). Source data are provided with this paper. All data is available from the corresponding author upon reasonable request. Source data are provided with this paper.

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

## Acknowledgements

The authors thank Michael Bustin for sharing HMGN1 antibody. Long read Iso-Seq was provided by the SMRT Grant of MDxK. This work was supported by the National Research Foundation of Korea (NRF-2020R1A2C3003352, NRF-2019R1C1C1008181, and NRF-2020R1C1C101220611) and the Institute for Basic Science (IBS-R008-D1). C.K. was supported by the KRIBB Research Initiative Programme. The authors thank Daisy Sunghee Lim for creating model images 1a, 3d, and 6.

## Author contributions

C.K. and S. Sung conducted most of experiments, analysed the data, and wrote the paper. J.-S.K. and S. Shin conducted quantitative proteomics analysis. H.L. conducted telomere-mTALT co-localisation experiment and RNaseH1-depletion experiments. Y.J. and J.J.S. conducted single-cell RNA-sequencing and analysed the result. E.K. conducted C-circle assay. J.K. conducted bioinformatics analysis of telomere repeat-containing reads. D.K. conducted SNP analysis. H.N. kindly provided the ALT mESC and supported the study. V.N.K. supervised the single cell RNA-sequencing. D.P. conducted most of bioinformatics analysis and supervised the manuscript. J.L. supervised the process of the experiments and wrote the paper.

## Competing interests

The authors declare no competing interests.
