## [Peer Review File · Nature Communications]

REVIEWER COMMENTS

Reviewer #1 (Remarks to the Author):

In this manuscript, Kim et al analyse the sequence content of the telomeres in a mESC ALT survivor cell line, which is known to contain non-telomeric sequences in its telomeric regions. The authors identify a subtelomeric sequence (mTALT) originating on chromosome 13, and later identified on chromosome 11, that becomes propagated in the telomeric regions of this cell line. This ALT cell line did not display the expected ALT phenotypes, and the authors attribute this to mechanistic diversity. The authors conduct interesting sequence analyses to determine the origin of the mTALT sequences, and to trace the amplification process, identifying the origin as being the chromosome 11 mTALT sequence that was duplicated prior to ALT activation. The authors then conduct analysis of the epigenetic state of the telomeres and characterise the presence of TERRA. Quantitative proteomics and transcriptomics identified HMGN1, which is known to compete with histone H1 for nucleosome binding, as being enriched in the ALT cells. HMGN1 was found to contribute to the ALT phenotypes, but it is unclear whether this gene was coincidentally duplicated and selected for. Overall, the figures are well presented, and it is an interesting study. I found the manuscript quite hard to follow, and the text requires substantial editing. The phenomenon of non-telomeric sequences being propagated in ALT telomeric regions is not totally novel, and I feel the interpretation that this may represent a new ALT cancer type is overstated.

Major points:

In the abstract, the authors state that "we show that mammalian telomeres can also be completely reconstituted using a non-telomeric unique sequence", whilst later they say that "canonical telomeric repeats were duplicated along with mTALTs". I don't think that it is correct that the mTALT sequence "constructs" new telomeres, I think it is more that the sequence becomes propagated through the telomeres. This should be clarified/discussed. I feel that statements eluding to a new ALT cancer type are not sufficiently supported by the data/phenotype. The presence of telomeric sequences within the mTALT sequences is also indicative of the cells activating ALT pathways, but mTALT sequences infiltrating the terminal regions and then becoming amplified.

A similar phenomenon has been reported previously in two papers published back to back in Cancer Research by Roger Reddel's and Brad Johnson's groups, in which SV40 sequences become propagated through human ALT telomeres (Fasching et al and Marciniak et al, 2005). Both studies report a reduction in ALT phenotypes, concomitant with the presence of aberrant sequences in the telomere regions. These studies detract from the novelty somewhat, and must be cited and discussed in the context of this work.

I didn't find the method for the C-circle assay. If the telomeres contain abundant mTALT sequences, they may still be forming extrachromosomal circular DNA, but the circles would not be detectable by the C-circle detection method. Sequence content should be taken into account with the ALT assays, and it would be interesting to see data using an mTALT probe.

Reviewer #2 (Remarks to the Author):

The manuscript „Mammalian telomeres reforged with non-telomeric sequences “ from Kim et al. describes the discovery of sequences in mouse that can function as template for alternative lengthening of telomeres. Using mouse cell lines to construct a time series that represent time points before and after the onset of ALT, the authors are able to monitor the progressive enrichment of these mTALT sequences at the ends of the mouse chromosomes, and track the origin of the sequence to one of two homologous sequences which represents the evolutionary younger copy. They furthermore use

the shift of copy number alteration profiles from a subclonal mix in the pre-ALT samples to a monoclonal architecture in the post-ALT samples to characterize the evolutionary bottleneck the initiation of ALT represents.

Overall the manuscript summarizes an impressive body of work that will drive the field forward, but several aspects of the text require further consideration.

Major comments:

- Have you provided a complete sequence of the mTALT elements in the supplementary data? If possible, do so to augment the following result: Line 141: "A terminal restriction fragment (TRF) assay revealed that only post-ALT cells showed discrete bands resulting from telomere non-cutter enzymes" Annotate the restriction sites in the sequence of mTALT and list the expected fragment sizes. Does these predictions match the some of the observed bands?

- You nicely used the SNP allele frequencies to identify the chr11 locus as the active locus. Beyond that, is the SNP data sufficient to decide if both alleles of the chr11 mTALT locus are used for ALT or only one allele? Be more explicit if you can clarify this point or if the alleles are too similar to make a conclusion.

- Line 227-228: "The amount of various repetitive sequences including rRNA did not significantly (Extended Data Fig. 3c)." Sentences is incomprehensible without the verb, and will change its meaning depending on which verb you will add. Completion necessary.

- Line 235: You indicate that "certain cells" were selected upon ALT activation. Do you have any reason to believe that this was not a monoclonal event happening in a single cell that then expanded? It would be helpful if you would discuss the possible alternatives in the light of the data you have gathered.

- In human cancer inactivation of the proteins ATRX and DAXX are frequently observed in association with ALT. You should briefly mention in one or two sentences if their mouse homologs are affected or not in your proteomics data. Atrx for instance is featured quite prominently in figure 5b, and appears to be as affected as Hmgn1, and is after FDR correction close to significance in the Volcano plot in Figure 5a.

- In line 152 and line 340 you mention the "right part" of the chromosomes 13 and 11, respectively. You may be more precise here using terms such as long arm / short arm and telomeric side / centromeric side etc.. In the current form it is easy to misunderstand these sentences.

- Line 362-364: "We showed that the actual ALT template sequence form heterochromatic telomere and promoted the transcription from insulator motif toward centromere." This sentences is incomplete and does not make sense in the current form. Reformulate!

- Does the fact that some mouse strains have the chr11 copy of mTALT while other have not impact the likelihood that the former evolve ALT compared to the later? Addressing this question in the Discussion may inspire follow-up analysis.

- For many of the computational tools used in the analysis you mention the versions, but do not cite the related scientific publications. This includes BWA-mem, Picard-Tools, Samtools, MACS2, DESeq2, Computel, TelomereHunter and Telomerecat. I would suggest that you carefully check for which of the third party software you have applied scientific publications exist which you then may cite in the manuscript.

- Figure 1 c in the Extended Data Appendix indicates that PD800/PD100 log₂ ratio shows an enrichment of TGAGGG and TTCGGG singleton repeats above the expected log₂ ratio. This is in contrast to the statement you make in line 146-149, where no differences are reported. Reformulate this part to match the observations you do report.

Minor comments:

- The first sentences of the abstract appears unprecise. How about: "Telomeres are part of a highly refined system for maintaining the stability of linear chromosomes."

- Line 76-78: "but ~15% of human cancer cells are known to maintain telomeres by a telomerase-independent telomere maintenance mechanism, which is called the alternative lengthening of telomeres (ALT)"

A more precise summary would be: "but ~15% of human cancer cells are known to maintain telomeres by telomerase-independent telomere maintenance mechanisms, which are summarized under the term alternative lengthening of telomeres (ALT)"

- Line 346-347: "telomere shortening dependent-chromosomal fusions" appears to be an odd use of the hyphen. Do you mean "telomere-shortening-dependent chromosomal fusions"?
- Sometimes upper and sometimes lower case is used to refer to figures. Decide for one form.
- In figure 5 the first panel is named aa instead of a.

Reviewer #3 (Remarks to the Author):

The authors provide evidence showing that, in mammals, telomeres can be reconstituted with a non-telomeric unique sequence (mTALT) used for repairing DNA damage at telomeres and for adding new telomeric sequences in ALT mESs. Using MS and epigenomics data before or after ALT activation, the authors show the expression of a non-coding mTALT transcript. After ALT activation, they find that HMGN1 protein levels increase and contribute to telomere stability through telomeric transcription. They propose that the non-canonical telomeric sequence has a functional role in cancer and evolution.

Overall, the data are interesting and for the most part the conclusions are supported from the observations shown. What is missing is further evidence on the functional role of HMGN1 in mTALT transcription and on R-loops formation as well as on the specificity of the findings to HMGN1 itself. The work would also benefit from the addition of few important controls as well as from a more detailed explanation on the proposed working model (in view of recent findings).

Specifically:

-the authors provide no evidence on the specificity of HMGN1 in telomere maintenance. This is important because their MS list (Fig 5) also includes other members of the HMG group i.e. HMGA, HMGN3. Would knocking down these protein targets lead to similar findings (e.g. R-loops, mTALT transcription, etc.)?

-is the effect of HMGN1 on mTALT transcription specific? Other targets (within and outside this region) should be included.

-On Figure 5L, the authors need to include an RNAaseH1 treatment to control for the presence of R-loops.

-The connection of HMGN1 to R loops (or in the resolution of R-loops) remains unclear in the present work. Is HMGN1 recruited to R-loops? An HMGN1 ChIP followed by DRIP could further highlight this point.

-Is depletion of HMGN1 leading to an increase in γ H2AX-associated DNA damage? If so, is the increased DNA damage levels due to R-loops? An RNaseH1 treatment in shHMGN1 cells could help answer this question.

-R-loops are known to be generated during abrupt changes in transcription demands or when transcription is blocked due to DNA damage. This should be discussed with respect to mTALT transcripts in shHMGN1 cells.

Other comments:

Figure 5b/Suppl. Figure 4a-c: The GO numbers of GO terms are not shown. Please provide the GO number for all GO terms shown.

Figure 5b: Is the heatmap shown referring to the number of peptides or is this a log2 fold change as shown in figure 5a? The term "count" should be explained in the figure legend.

Figure 5d: the authors should validate the protein levels of a few more mass spec targets (beyond HMGN1). It is not entirely clear why the authors have chosen to focus on HMGN1 alone or why they have neglected other factors e.g. HMGA or HMGN3 in the MS list that seem equally relevant.

Figure 5f: do the three independent replicates mentioned represent biological or technical replicates? This information should be provided here and elsewhere in the manuscript.
Figure 6: please provide a legend that summarizes what is shown in the figure.

Responses to reviewers' comments

Reviewer #1

Q: Overall, the figures are well presented, and it is an interesting study. I found the manuscript quite hard to follow, and the text requires substantial editing. The phenomenon of non-telomeric sequences being propagated in ALT telomeric regions is not totally novel, and I feel the interpretation that this may represent a new ALT cancer type is overstated.

A: We agree with the reviewer's comment. We have extensively revised the manuscript and employed a professional editing company (ENAGO) to edit our manuscript. We also toned down our discussion on the possibility of a new ALT cancer type. A more detailed description of our revision is provided below.

Major points:

Q1. In the abstract, the authors state that "we show that mammalian telomeres can also be completely reconstituted using a non-telomeric unique sequence", whilst later they say that "canonical telomeric repeats were duplicated along with mTALTs". I don't think that it is correct that the mTALT sequence "constructs" new telomeres, I think it is more that the sequence becomes propagated through the telomeres. This should be clarified/discussed. I feel that statements eluding to a new ALT cancer type are not sufficiently supported by the data/phenotype. The presence of telomeric sequences within the mTALT sequences is also indicative of the cells activating ALT pathways, but mTALT sequences infiltrating the terminal regions and then becoming amplified.

A1. Thank you for your constructive comments. As you pointed out, telomeres of ALT mESCs do contain telomeric repeats as well as mTALT sequences. We did not intend to present mTALT as the sole component of ALT telomeres, but some expressions we used would be confusing to readers. As you adequately pointed out, mTALTs were replicated with telomeric repeats, so we agree that the expression of 'construct' may be misleading. Therefore, we replaced several expressions with alternatives to explain mTALT as a constituent of telomere contents.

We revised the manuscript as follows:

(Page 2, line 47) *Here, we show that mammalian telomeres can also be completely reconstituted using a non-telomeric unique sequence.*

→ (line 43) **Here we show that mammalian telomeres can also exploit non-telomeric, unique sequences in addition to telomeric repeats.**

(Page 5, line 121) *Our findings suggest that the robust telomere system based on simple repeats can be replaced with the ALT mechanism reconstructing telomere with new sequences even in mammalian cells.*

→ (line 115) **Our findings suggest that the robust telomere system based on simple repeats can convert to the ALT mechanism making use of non-telomeric sequences even in mammalian cells.**

(Page 7, line 174) *To examine how mTALT constructs new telomeres at a sequence level, we analyzed telomere-adjacent sequences using paired-end WGS reads.*

→ (line 162) **To examine how mTALT constitutes new telomeres at a sequence level, we analysed telomere-adjacent sequences by using paired-end WGS reads.**

(Page 10, line 242) *To determine the epigenetic state of the newly constructed telomeres with mTALT,*

→ (line 243) **To determine the epigenetic state of the mTALT-containing ALT telomeres,**

(Page 12, line 302) *This suggests that the gene duplication of Hmgn1 may have been coincidentally selected with ALT activity in which telomeres are reconstructed with mTALTs.*

→ (line 322) **This suggests that the gene duplication of Hmgn1 may have been coincidentally selected with ALT activity.**

(Page 13, line 317) *In this study, we established a mESC ALT model in which telomeres are constructed and maintained by the non-telomeric unique sequence mTALT.*

→ (line 351) **In this study, we established an mESC ALT model in which the non-telomeric unique sequence, mTALT, was utilised for telomere maintenance.**

(Page 13, line 325) *In various species, specific sequences have been known to reconstruct telomeres.*

→ (line 359) **In various species, specific sequences have been known to change telomere composition.**

(Page 13, line 328) *This is the first description of a specific mammalian ALT template that has the ability to protect telomere even when telomerase activity exists and eventually reconstruct telomeres.*

→ (line 362) **This is the first description of a specific mammalian ALT template that can protect telomeres even when telomerase activity exists and that can eventually be incorporated into telomeres.**

An interesting question that was raised during this research was whether mTALT was randomly (or accidentally) mobilized to be included in telomeres in ALT cells. We would like to think that mTALT is not just a random sequence, but a sequence that bears some characteristics to be easily recruited to telomeres. Underlying our thought is the fact that mTALT has been selected as a telomere-protecting sequence at least twice. One is the process of repairing telomeric damage in the 129/Ola strain, which is manifested by the duplication of mTALT from chromosome 13 to chromosome 11, and the other is the ALT activation process in the ALT survivors that emerged in the telomerase-deficient ES cells of the 129/Ola origin. Thus, we think that mTALT has some characteristics related to the propensity to be recruited to telomeres. The authors hope that the reviewer agrees with our thoughts.

Q2. A similar phenomenon has been reported previously in two papers published back to back in Cancer Research by Roger Reddel's and Brad Johnson's groups, in which SV40 sequences become propagated through human ALT telomeres (Fasching et al and Marciniak et al, 2005). Both studies report a reduction in ALT phenotypes, concomitant with the presence of aberrant sequences in the telomere regions. These studies detract from the novelty somewhat, and must be cited and discussed in the context of this work.

A2. Thanks for the great suggestion. We added the following sentences in the discussion section.

(line 370) "In humans, tags flanked by telomeric repeats are reported to have the potential to be duplicated to other chromosomes²⁸. For example, the SV40 sequence can be integrated into telomeres during immortalisation of the Werner syndrome cell line^{29,30}. It is notable that SV40-based ALT cells also did not show the common ALT marker, APB, which suggests a similar mechanism of ALT to that of ALT mESCs we

report here. One difference between the previously reported cases and our findings is that the mTALT sequence we described naturally exists in the mouse genome as a source of templates for ALT telomeres, which underscores its cell-intrinsic potential of reshaping telomeres.”

Q3. I didn't find the method for the C-circle assay. If the telomeres contain abundant mTALT sequences, they may still be forming extrachromosomal circular DNA, but the circles would not be detectable by the C-circle detection method. Sequence content should be taken into account with the ALT assays, and it would be interesting to see data using an mTALT probe.

A3. Thank you for the constructive suggestion. We performed C-circle assays without any restriction enzymes. In the revision, to be more precise, we repeated the C-circle assay with mTALT-specific probes as the reviewer suggested (New Extended Data Fig. 1e). We also added a positive control, U2OS, which was reported to bear a large number of C-circles. Consistently, we did not find any difference of C-circle between PD100 cells and PD800 cells.

We also described the details of the method of the C-circle assay as follows.

(line 512) 'C-circle assay was done based on previously reported way with a few modifications⁴⁰. We did not use restriction enzymes and Exonuclease to preserve a potential mTALT-based C-circles. Two kinds of probe were used to detect C-circle signals. One was the telomeric probe dig-(CCCTAA)*4, and the other was the mixture

of mTALT specific probes which were established with DIG DNA labeling mix (Roche) according to manufacturer's guides. Primers used for making mTALT specific probes were listed. probe 1F: CCAAGCAAGTAGCAGAGATTAGC, probe 1R: TCTACACACACTTACAGCCCTGA, probe 2F: TTCTACATCCCCGCACACAC, probe 2R: AGGCGAGGGGTAAGAAGAGA, probe 3F: GATCCCAGATTCTGCTAAGACT, probe 3R: GTCCTGTTTCTAGGGGTGAGTTC. Φ 29 polymerase-amplified reactants were blotted onto nitrocellulose membrane with slot-blotter and visualized with DIG luminescent detection kit (Roche) according to manufacturer's guides.'

Reviewer #2

Major comments:

Q1. Have you provided a complete sequence of the mTALT elements in the supplementary data? If possible, do so to augment the following result: Line 141: "A terminal restriction fragment (TRF) assay revealed that only post-ALT cells showed discrete bands resulting from telomere non-cutter enzymes" Annotate the restriction sites in the sequence of mTALT and list the expected fragment sizes. Do these predictions match some of the observed bands?

A1. Thank you for the suggestion. We added a complete sequence of the mTALT in Extended Data Fig. 3. We used two sets of restriction enzymes to perform a TRF assay. 1) AluI/MboI and 2) HinfI. In the case of the AluI/MboI combination, there are 77 cutting sites inside mTALT (fragment size average: 93, min: 3, max: 357). In the HinfI case, 34 cutting sites are present inside mTALT (fragment size average: 208, min: 11, max: 582). In TRF assay, we used a telomeric repeat-specific probe so we could detect the length of fragments of telomeric repeats present between two mTALTs. We can predict the length of mTALT fragments remaining at the 5' and 3' terminal regions of

mTALT after restrictions. In the case of AluI/MboI the length of the remaining mTALT is 150 bps. For HinfI, the length is 880 bps. So, we can predict the difference of each TRF signal will be 730 bps. In practice, the major TRF size of AluI/MboI case is ~1.2 kbps and that of HinfI is ~1.9 kbps. From the result, we can also find that the length of telomeric repeats between two mTALTs is ~1 kbps. The result was also confirmed by *in silico* digestion. The figure below is attached to Extended Data Fig. 2. The matched explanation appears in line 187 of the revised manuscript.

Q2. You nicely used the SNP allele frequencies to identify the chr11 locus as the active locus. Beyond that, is the SNP data sufficient to decide if both alleles of the chr11 mTALT locus are used for ALT or only one allele? Be more explicit if you can clarify this point or if the alleles are too similar to make a conclusion.

A2. Thank you very much for your suggestion. That is a very interesting question, unfortunately, inbred laboratory mouse strains are characterized by at least 20 generations of inbreeding and are genetically homozygous at almost all loci (Beck et al, Nature genetics, 2000). However, heterozygous SNPs were detected when sequencing reads of 16 mouse strains were aligned to the C57BL/6J reference genome. These heterozygous SNPs mainly originated from copy number change or novel paralog formation (Lilue et al., Nature genetics, 2018). In the case of our mTALT, the heterozygous allele observed in the sequencing data is due to the presence of two copies of mTALT. To check the interesting possibility that you proposed, we amplified the specific chr11 mTALT region with PCR, after which we sequenced the chr11 mTALT amplicons using MiSeq. The result was that the allele frequencies of all mTALT regions were over 90%. Therefore, no discernible alleles existed in the chr11 mTALT region so we cannot determine which strand was used as the ALT template. The figure below is

attached to Extended Data Fig. 2. The matched explanation appears in line 188 of the revised manuscript.

Chr. 11 mTALT PCR analysis.

Allele frequency of Chr.11 amplicon

Q3. Line 227-228: "The amount of various repetitive sequences including rRNA did not significantly (Extended Data Fig. 3c)." Sentences is incomprehensible without the verb, and will change its meaning depending on which verb you will add. Completion necessary.

A3. Sorry for the mistake. We added the verb ‘change’.

Q4. Line 235: You indicate that “certain cells” were selected upon ALT activation. Do you have any reason to believe that this was not a monoclonal event happening in a single cell that then expanded? It would be helpful if you would discuss the possible alternatives in the light of the data you have gathered.

A4. We thank the reviewer for the point. As the reviewer pointed out, we cannot rule out the possibility that ALT was activated in a single cell, and then the cell was diversified during expansion. To trace the clonal evolution and lineage, it would be necessary to explore longitudinal whole genome sequencing data at a single-cell level, which we have not gathered for this study yet. Instead, we have produced the bulk WGS data and investigated the change in copy number variation (CNV), showing that highly heterogeneous pre-ALT PD350 cells underwent selection to become homogeneous post-ALT PD450 cells (Figure 3d). Tens of millions of cells were under telomeric crisis with genomic aberrations at PD350, and a certain proportion of the cells would share the genomic ALT potentials within the population during cellular replication. Based on these, we speculated that ALT was highly likely to be multi-cellular events even though the events would not simultaneously happen. However, we agree that ALT activation in multiple cells was not fully supported by the current data. Therefore, as suggested by the reviewer, we have added the possible alternatives in the Discussion section. Also, we have changed ‘certain cells’ into ‘certain cells or a single cell’ in Line 233. We added a paragraph explaining the hypothesis in the Discussion section as follows:

(line 393) ‘The analysis of CNVs using bulk WGS data showed that highly heterogeneous pre-ALT PD350 cells underwent selection to become homogeneous post-ALT PD450 cells (Figure 3d). At PD350, tens of millions of cells were under telomeric crisis with genomic aberrations, and a certain proportion of the cells would share ALT potentials. The selection upon ALT activation was likely to be multi-cellular events even though the events would not simultaneously happen. However, the genomic structure at PD450 and PD800 also can be shaped from monoclonal ALT activation followed by the clonal expansion, so bulk WGS data have limitations in revealing the clonal evolution at the level of single cells after PD350 upon ALT activation. Therefore, further studies using longitudinal single-cell WGS data will be necessary to trace the mono- or

polyclonal evolution and lineage of ALT-activated cells.’

Q5. In human cancer inactivation of the proteins ATRX and DAXX are frequently observed in association with ALT. You should briefly mention in one or two sentences if their mouse homologs are affected or not in your proteomics data. Atrx for instance is featured quite prominently in figure 5b, and appears to be as affected as Hmgn1, and is after FDR correction close to significance in the Volcano plot in Figure 5a.

A5. Thank you for your suggestion. The increase of ATRX was an unexpected result to us as well, and we did not have any insight on how to explain the result. Now, we think that the observation of increased ATRX indicates that our case is different from the previously reported ALT cases. Therefore, thanks to the reviewer’s suggestion, we added two sentences at line 304 to briefly introduce the result as follows:

‘Meanwhile, human ALT cancers are highly correlated with the inactivation of ATRX and DAXX proteins ¹⁷. Interestingly, the expression of ATRX was increased, rather than decreased, in post-ALT mESCs, and the expression of DAXX was not affected by ALT activation, which implicates a different mode of action of ALT in mESCs’.

Q6. In line 152 and line 340 you mention the “right part” of the chromosomes 13 and 11, respectively. You may be more precise here using terms such as long arm / short arm and telomeric side / centromeric side etc.. In the current form it is easy to misunderstand these sentences.

A6. Thank you for the suggestion. We changed the expression ‘right part’ to ‘the long arm’.

Q7. Line 362-364: “We showed that the actual ALT template sequence form heterochromatic telomere and promoted the transcription from insulator motif toward centromere.” This sentence is incomplete and does not make sense in the current form. Reformulate!

A7. Thank you for your comment. The pointed sentence is somehow overlapped with the first sentence in the paragraph. We formulated the original sentence as a brief and precise one:

(line 429) ‘We showed that the mTALT-based telomeres also appeared as heterochromatic, and at the same time, produced a number of transcripts.’

Q8. Does the fact that some mouse strains have the chr11 copy of mTALT while other have not impact the likelihood that the former evolve ALT compared to the later? Addressing this question in the Discussion may inspire follow-up analysis.

A8. Thank you very much for bringing up an interesting point here. We discussed the impact of copy numbers of mTALT onto the ALT activation in our draft, but during the process of polishing our manuscript somehow that part has been omitted. We added back a paragraph to discuss the topic adequately as follows:

(line 403) ‘It is noteworthy that there is a natural variation in the copy number of mTALT. During telomere shortening, genomic heterogeneity was increased in terms of copy numbers. mTALT may have gone through similar copy-number heterogeneity around senescence and cells containing several mTALTs may have a higher chance of being selected in ALT activation. The naturally increased copy number of mTALT in some strains may be a sign of repairing telomere damage which could not be handled by telomerase. In the Hawaiian strain of *C. elegans*, the duplication of TALT at the telomeric end was a prerequisite for the introduction of TALT-mediated ALT. Our finding of ALT mESC may have been possible due to starting with a specific strain bearing the right template at the right place.’

Q9. For many of the computational tools used in the analysis you mention the versions, but do not cite the related scientific publications. This includes BWA-mem, Picard-Tools, Samtools, MACS2, DESeq2, Computel, TelomereHunter and Telomerecat. I would suggest that you carefully check for which of the third party software you have applied scientific publications exist which you then may cite in the manuscript.

A9. Thank you for your critical comment. We added all citations for tools we used.

Q10. Figure 1 c in the Extended Data Appendix indicates that PD800/PD100 log₂ ratio shows an enrichment of TGAGGG and TTCGGG singleton repeats above the expected log₂ ratio. This is in contrast to the statement you make in line 146-149, where no differences are reported. Reformulate this part to match the observations you do report.

A10. As the reviewer pointed out, although TGAGGG and TTCGGG increased in PD800, they increased less than common ALT cancer cell lines or ALT patients. To avoid confusion, we compared the typical ALT cell lines derived from the WI-38 primary cell with HeLa, the telomerase positive cancer cell lines as a positive control. Also, to avoid confusion, we have specified the representative ALT variant repeats that are significantly changed in various ALT cancers in the main text of the revision (line 139).

‘Molecular markers of ALT cancers, such as variant telomeric repeats (e.g. TCAGGG and TGAGGG), C-circle (extrachromosomal telomeric DNA circle of C-rich strand), and APBs (ALT-associated promyelocytic leukaemia bodies), were not different between pre-ALT and post-ALT cells (Extended Data Fig. 1b–e)^{6,15}, increasing the likelihood that ALT mESCs have different characteristics from known ALT cancer models.’

The figures below are attached to Extended Data Fig. 1.

Minor comments

Q11. The first sentences of the abstract appear unprecise. How about: “Telomeres are part of a highly refined system for maintaining the stability of linear chromosomes.”

A11. Thank you for your suggestion. We changed the sentence as you suggested. (line 38)

Q12. Line 76-78: “but ~15% of human cancer cells are known to maintain telomeres by a telomerase-independent telomere maintenance mechanism, which is called the alternative lengthening of telomeres (ALT)” A more precise summery would be: “but ~15% of human cancer cells are known to maintain telomeres by telomerase-independent telomere maintenance mechanisms, which are summarized under the term alternative lengthening of telomeres (ALT)”

A12. Thank you for your suggestion. We changed the sentence as the reviewer suggested. (line 82)

Q13. Line 346-347: “telomere shortening dependent-chromosomal fusions” appears to be an odd use of the hyphen. Do you mean “telomere-shortening-dependent chromosomal fusions”?

A13. Thank you for your suggestion. We changed the sentence as you suggested. (line 414)

Q14. Sometimes upper and sometimes lower case is used to refer to figures. Decide for one form. In figure 5 the first panel is named aa instead of a.

A14. We fixed the mistakes.

Reviewer #3

Q1. the authors provide no evidence on the specificity of HMGN1 in telomere maintenance. This is important because their MS list (Fig 5) also includes other members of the HMG group i.e. HMGA, HMGN3. Would knocking down these protein targets lead to similar findings (e.g. R-loops, mTALT transcription, etc.)?

A1. Thank you very much for your suggestions. We selected HMGN1 as the main candidate for ALT regulation based on our single-cell RNA sequencing data as well as

the proteomics results. As the reviewer pointed out, there are several candidate genes to be tested if the candidates are selected depending solely either on proteomics or transcriptomics data, but we took several criteria into consideration. When we ranked genes based on the degree of their impacts in differentiating post-ALT (PD800) cells from pre-ALT (PD100) cells, HMGN1 was one of the genes highly ranked. Other genes that showed higher rank in the list were not identified in other experiments such as bulk RNA-seq, proteomics, and ATAC-seq. Moreover, none of those genes with higher rank in the list did not have any sequence information regarding telomere regulation. Because HMGN1 was placed at the top in both proteomics and single-cell RNA-sequencing, we had to pick this gene as the first candidate to examine.

On the other hand, other HMG genes also seem to have the potential to regulate telomere maintenance. Several studies identified HMG proteins as telomere-binding proteins, shelterin-binding proteins or TERRA-interacting proteins. Therefore, it is reasonable to investigate their roles in ALT. We depleted HMGA1, HMGA2 and HMGN3 using lentivirus based shRNAs and assessed their effects on telomere damage, mTALT transcription and telomere length. Telomeric damage was induced with all HMG-depleted cases. HMGA1 depletion induced a decrease in TERRA expression, but the effect of HMGN1 depletion was more severe and consistent. HMGA2 or HMGN3 depletion increased mTALT transcription with one primer set. The depletion of other HMG genes did not show significant alterations in telomere length as HMGN1-depletion. Taking all results into account, we concluded that HMGN1 is the most influential factor for telomere physiology, especially for telomere length. The interaction between other HMG proteins and telomeres will be our next area to expand on. The figures below are attached to Extended Data Fig. 8. The matched explanation appears in line 332 of the revised manuscript.

C

Q2. is the effect of HMGN1 on mTALT transcription specific? Other targets (within and outside this region) should be included.

A2. Thank you for the comment. HMGN1 does not have enzymatic activity on its own, so the resultant effect on chromatin may differ concerning interacting partners. As HMGN1 is a global chromatin protein, we do not think HMGN1 has only telomere-specific effects. However, as shown in our data, HMGN1 localizes to telomeres and has a protective role for telomeres. Another study showed HMGN1 interacted with a major shelterin protein, TRF2 (Ok-Hee et al., 2011). Thus, it will be a fair statement that HMGN1 may affect transcriptional networks in a genome-wide way, but it also regulates telomere physiology.

To examine these points in the most stringent way, we performed RNA-seq with control PD800 cells and HMGN1-depleted PD800 cells during revision. When the differentially expressed genes were sorted according to their q-value (adjusted p-value), Hmgn1 was the most significantly decreased gene. Notably, the next significantly decreased transcript was mTALT, which suggests HMGN1 regulates mTALT

transcription critically. Another interesting observation was that the expression of *Tcstv3*, the gene located inside mTALT region, was increased. HMGN1 seems to affect the directionality of transcription because *Tcstv3* is transcribed to the opposite direction of mTALT. We put this result in main figure 5i.

Notably, no evidence was found that HMGN1 regulates regions adjacent mTALT specifically or regulates genes related to ALT maintenance. First, when *Hmgn1* was knocked down, genes close to mTALT (such as *Ptchd3*, *Metrn1*, *B3gnt1*, and *Zfp750*) did not change specifically (refer to the volcano plot below). Many other genes including those on chromosome11 (refer to the dot plot below) were affected by HMGN1 depletion. Second, when examining the Gene Ontology (GO) of the genes that were downregulated under *hmgn1* knockdown, GO terms included multicellular organism development, positive regulation of cell proliferation, cell adhesion, and a few unrelated to terms. These terms are not likely to be directly related to ALT mechanism. Therefore, we think that HMGN1 is involved in the ALT mechanism by regulating mTALT transcription in a strand-specific manner. Although we cannot fully exclude the possibility that HMGN1 regulates ALT-related genes, we could not find ALT-specific terms with RNA-seq of HMGN1-depleted cells.

GO Term	Count	%	P-value	FDR
Multicellular organism development (GO:0007275)	42	14.40%	3.40E-10	5.70E-07
Positive regulation of cell proliferation (GO:0008284)	26	8.90%	8.60E-08	7.10E-05
Nervous system development (GO:0007399)	21	7.20%	2.00E-07	1.10E-04
Cell adhesion (GO:0003723)	23	7.90%	7.00E-07	2.90E-04
Axon guidance (GO:0007411)	12	4.10%	5.60E-06	1.90E-03
Neuron differentiation (GO:0030182)	11	3.80%	8.40E-06	2.30E-03
Transcription, DNA-templated (GO:0006351)	49	16.80%	9.80E-06	2.30E-03
Positive regulation of transcription from RNA polymerase II promoter (GO:0045944)	32	11.00%	1.10E-05	2.30E-03
Circadian rhythm (GO:0007623)	10	3.40%	1.50E-05	2.30E-03
Negative regulation of transcription from RNA polymerase II promoter (GO:0000122)	26	8.90%	1.80E-05	2.50E-03
Positive regulation of cell migration (GO:0030335)	13	4.50%	2.00E-05	2.60E-03
Cell fate commitment (GO:0045165)	8	2.70%	5.10E-05	5.60E-03
Cell-cell signaling (GO:0007267)	9	3.10%	7.50E-05	7.80E-03

Q3. On Figure 5L, the authors need to include an RNAaseH1 treatment to control for the presence of R-loops.

A3. Sorry for the confusion that we may have caused by using abbreviations without definition. ‘RH’ in the figure was RNaseH1 treatment control. To reduce the potential confusion, we changed ‘RH’ to ‘RNaseH1’.

Q4. The connection of HMGNI to R loops (or in the resolution of R-loops) remains unclear in the present work. Is HMGNI recruited to R-loops? An HMGNI ChIP followed by DRIP could further highlight this point.

A4. Thank you for the constructive comment.

Basically, what was known about HMGNI was that it loosens the compact structure of chromatin by competing out linker histone H1 and recruiting other chromatin remodelers. Our data showed that HMGNI enhanced mTALT transcription and the increased telomeric transcripts in turn formed R-loops. This phenomenon seems to be a result of the chromatin changes induced by the high expression of HMGNI. Because HMGNI functions as a scaffold to recruit other proteins, we tend to think that the connection of HMGNI to R loops is an indirect one in which the consequence of HMGNI function is R loop formation at the telomeres. We do not know yet which

HMGN1-interacting factors have enzymatic roles in the chromatin opening process and R-loop formation. It can be interesting to study the role of HMGN1-interacting proteins in telomere and ALT.

As suggested by the reviewer, it will be interesting to examine whether HMGN1 is directly recruited to R loops. Unfortunately, it is too hard for us to perform CHIP and DRIP in the same sample, so we chose an alternative way. To observe the interaction of HMGN1 and R-loop, we performed proximity ligation assay (PLA), which can capture protein-protein interaction, in this case, the interaction of anti-HMGN1 antibody and S9.6 R-loop specific antibody. We compared three cells, pre-ALT PD100, post-ALT PD800 and HMGN1-depleted PD800 (which have lowered HMGN1 and R-loop in telomeres). PD800 cells showed more PLA signals than PD100 cells as expected. HMGN1 depletion lowered PLA signals in PD800 cells. These data indicate that HMGN1 and R-loop were located close enough to be detected by PLA, and PLA signals were sufficiently specific.

We do not have enough data to build a concrete model for the process by which HMGN1 is recruited to R-loop. However, some reports show that transcription-coupled nucleotide excision repair (TC-NER) factors may act to process R-loops, leading to DNA breaks (Shivji et al., 2018; Sollier et al., 2014; Yasuhara et al., 2018). HMGN1 is a member of TC-NER factors so HMGN1 may be recruited to R-loop for processing. Thus, HMGN1 and R-loop may have a complicated inter-connectivity that HMGN1 leads to R-loop formation as well as R-loop recruits HMGN1. It will be a very interesting topic for further study. However, we want to streamline the logical flow of

this paper for readers' understanding, especially focusing on the function of HMGN1 onto R-loop formation. So please understand the situation in which we deliver the data above solely to reviewers.

Q5. Is depletion of HMGN1 leading to an increase in γ H2AX-associated DNA damage? If so, is the increased DNA damage levels due to R-loops? An RNaseH1 treatment in shHMGN1 cells could help answer this question.

A5. Thank you for the comment. Let us clarify what we have found about the effects of HMGN1 depletion on DNA damage and R loop formation. First of all, we presented the effect of HMGN1 depletion on DNA damage in Figures 5i and Extended Data Fig. 8c. As the reviewer pointed out, HMGN1 depletion increased telomeric DNA damage. However, we found that HMGN1 depletion also lowered TERRA expression and R-loop formation. Thus, it is unlikely that increased DNA damage was due to R-loops as suggested by the reviewer. Instead, we interpreted these results as TERRA/R-loop executed a protective role in mTALT-based telomeres.

During revision, we additionally experimented to test the effect of RNaseH enzyme on telomere protection (Extended data Fig. 8g). We used a lentivirus-based shRNA construct to deplete RNaseH1, and found that the lowered RNaseH induced the increased telomeric DNA damage. In this case, the telomeric R-loop formation was increased. Based on these findings, we can say that an elaborate balance of R-loop is important for stable telomere maintenance. Excess or too little amounts of R-loop may provoke problems in telomeres. We added a paragraph in discussion to elaborate these notions as follows (line 449):

'R-loops can induce genome instability by interfering with transcription and replication. In particular, DSB formation and DNA loss may occur if a fork collapse occurs following replication fork stalling. However, several reports allude to potential R-loop contribution to the maintenance of the genome, particularly the telomere. This is possible by chromatin regulation ³⁴, priming DNA replication ³⁵ and promotion of inter-telomere HR ³⁶. Of note, R-loops can prevent telomeric replication fork collapse through HR which prevents telomeres from becoming dysfunctional ³⁷. In a study of ALT cancer cells, when the amount of RNaseH1 was depleted or overexpressed, abrupt

telomere shortening occurred ³⁸. In other words, the precisely controlled telomeric R-loop makes an important contribution to telomere stability without seriously harming telomeric integrity.'

Q6. R-loops are known to be generated during abrupt changes in transcription demands or when transcription is blocked due to DNA damage. This should be discussed with respect to mTALT transcripts in shHMGN1 cells.

A6. Thank you for the suggestion. We did not provide an adequate explanation for the formation process of R-loops. Following your comments, we added a paragraph in the discussion as follows (line 435):

'R-loops can be generated during abrupt changes in transcriptional demands or when transcription is blocked ²⁰. The formation of telomeric R-loops of ALT mESC can also be explained by the chromatin decompaction effect produced by HMGN1 which increases telomere transcription, and the loosened chromatin structure which may assist the interaction of transcribed RNA and DNA. In addition, the increased telomere damage level after ALT activation may interfere with transcription progress and promote the generation of co-transcriptional R-loops. Reasons for the increased telomere damage after ALT activation could not be accurately determined but it may be due to the lowered shelterin density which promotes stable telomere replication and protection. Another interesting aspect is that the formation of R-loops can be increased when there is a conflict between DNA replication and transcription machinery (so-called transcription-replication collisions, TRCs) ³³. While telomere replication proceeds from the subtelomere to the end of the chromosome, the direction of mTALT

transcription is the reverse. Head-on TRCs specifically promote R-loop formation. Thus, the association between the mTALT sequence characteristics and HMGN1 function produced telomeric R-loops and regulated telomere physiology.

Other comments

Q7. Figure 5b/Suppl. Figure 4a-c: The GO numbers of GO terms are not shown. Please provide the GO number for all GO terms shown.

A7. We provided the appropriate GO numbers in the main and supplementary figures.

Q8. Figure 5b: Is the heatmap shown referring to the number of peptides or is this a log₂ fold change as shown in figure 5a? The term “count” should be explained in the figure legend.

A8. We appreciate the point and agree that the term was ambiguous. The heatmap is based on Z-scores of the relative quantity of proteins measured by LC-MS/MS analysis (relative report ion intensity) in log₂ scale, not the number of peptides. In colour index, ‘value’ indicates z-score of protein expressions and ‘count’ (with regard to histogram) indicates the accumulated number of indicated values represented in the heatmap. For better understanding, we replaced ‘value’ with ‘z-score’. We also added the explanations to figure legend (line 1016).

Q9. Figure 5d: the authors should validate the protein levels of a few more mass spec targets (beyond HMGN1). It is not entirely clear why the authors have chosen to focus on HMGN1 alone or why they have neglected other factors e.g. HMGA or HMGN3 in the MS list that seem equally relevant.

A9. We added our explanations and data at the related major comment Q1. We believe that the reviewer requested the orthogonal validation of protein level for a few more mass spec targets since the reviewer might have a doubt in the credibility of the protein quantification accuracy from proteomics result.

In fact, we adopted the most advanced proteome quantification method and a state-of-art mass spectrometer, i.e., TMT-labeling & synchronous precursor selection (SPS)-MS3 method. The protein quantification accuracy of TMT-labeling/SPS-MS3 method has been robustly demonstrated {ref 1-4} and popularly being used for a number of biological studies. In this regard, we did not try to perform more western blot

experiments on the purpose of protein level validation for other proteins except HMG1, the featured protein in our study. Although we agree with the reviewer that the orthogonal validation of protein expression levels can be beneficial in general, in this time we think the additional validation would not be necessary considering the quantitative accuracy level of the proteomics method in our study. Thus, we added only experimental results regarding how other HMG genes affected telomere physiology.

Ref

1. **MS3 eliminates ratio distortion in isobaric multiplexed quantitative proteomics. Nature Methods 8, 937–940 (2011)**
2. **MultiNotch MS3 Enables Accurate, Sensitive, and Multiplexed Detection of Differential Expression across Cancer Cell Line Proteomes. Anal. Chem. 86, 14, 7150–7158 (2014)**
3. **A Triple Knockout (TKO) Proteomics Standard for Diagnosing Ion Interference in Isobaric Labeling Experiments. J. Am. Soc. Mass Spectrom. 27, 10, 1620–1625 (2016)**
4. **TKO6: A Peptide Standard To Assess Interference for Unit-Resolved Isobaric Labeling Platforms. J. Proteome Res. 18, 1, 565–570 (2019)**

Q10. Figure 5f: do the three independent replicates mentioned represent biological or technical replicates? This information should be provided here and elsewhere in the manuscript.

A10. Thank you for your suggestion. We modified the relevant sentences in legends of 1b, 4i, 4k, 5g, 5m and 5o as ‘biologically independent replicates’.

Q11. Figure 6: please provide a legend that summarizes what is shown in the figure.

A11. Thank you for your helpful advice. We changed the figure title to a clear one and added legend contents for readers to understand the content easily as follows:

‘Fig. 6. A working model of ALT activation of mESCs. This figure summarises the emergence process of ALT mESC from various aspects. After the Terc gene was knocked out, cells of the early generation grew normally, but as the telomere length decreased growth slowed down and cells approached a senescent state. Affected by telomere shortening, genome-wide copy-number changes and chromosomal fusions

occurred, and transcriptional networks and cellular physiology changed accordingly. One or more cells activated ALT and were selected to form a homogenous post-ALT cell population. In an evolutionary time-line, the mTALT sequence was located at chromosome 13 first and a point replicated at the end of chromosome 11. In the process of telomere shortening, the only mTALT of chromosome 11 underwent selective duplication, and during ALT activation, the mTALT of chromosome 11 was amplified in cis and trans to cover all telomeres. Through transcriptome and proteome analysis, we confirmed that genome-wide epigenetic remodelling occurred during the ALT activation process, and the expression of telomeric transcripts and R-loop formation increased due to HMGN1-dependent telomeric changes, contributing to telomere maintenance.'

REVIEWER COMMENTS

Reviewer #1 (Remarks to the Author):

The authors have written a very nice rebuttal and have done a good job in addressing the reviewers' comments. The manuscript is substantially improved.

I still have some problems with the manuscript, mostly relating to the idea that the ALT mechanism can be "converted" to using non-telomeric sequences, and that this represents a "backup telomere maintenance mechanism capable of producing new telomeres" (taken from the Introduction). I don't find this explanation compelling. It seems more likely that sequences that are amenable to heterochromatinization can infiltrate telomeres and contribute to telomere capping function. I think this could be more clearly articulated.

I also have a problem with the novelty. First, this study is in mice, in which the significance, utility, mechanistic characterization and relevance of ALT is not clear. Second, a similar process has been observed in human ALT cells (as mentioned in my previous review). Although, these back to back studies have now been mentioned in the Discussion, they are not discussed fully or described in the Introduction, which I feel is important to set the scene for this work.

In the abstract, please state ALT as alternative lengthening of telomeres (not alternative telomere lengthening), telomere loss does not directly induce telomere recombination.

In the Introduction, please correct the first sentence, change the phrase "telomeres can be "broken" by cellular stresses".

I'm also not convinced by the telomere analysis in Fig 1, for instance in Fig 1j are we not supposed to be seeing TTAGGG repeats being replaced by mTALT from PD100 to PD800? I cannot see this in the figure. Also, have these cell lines been STR profiled for authenticity? I can't find this in the methods. The chromosomes look very different at PD100 compared to PD800.

Reviewer #2 (Remarks to the Author):

My comments have been adequately addressed.

Reviewer #3 (Remarks to the Author):

The authors have provided a wealth of information in the revised version, which I think merit publication.

Q: I still have some problems with the manuscript, mostly relating to the idea that the ALT mechanism can be "converted" to using non-telomeric sequences, and that this represents a "backup telomere maintenance mechanism capable of producing new telomeres" (taken from the Introduction). I don't find this explanation compelling. It seems more likely that sequences that are amenable to heterochromatinization can infiltrate telomeres and contribute to telomere capping function. I think this could be more clearly articulated.

A: We agree with the reviewer because our expression that the ALT mESC will use a new ALT mechanism that is different from previously known ALT mechanisms is not accurate, or supported by experimental evidence. It will be an important task for us to define what mechanism ALT mESC uses in our future studies. We discussed about the heterochromatic feature of mTALT in the discussion section, but, as the reviewer properly pointed out, we did not clarify how this feature actually affected the telomere protection function. Therefore, in our second revision, we simply explain the situation in which non-telomeric sequences co-exist with telomeric repeats and propose that this ALT phenomenon may be well conserved.

While we totally agree with the reviewer, we still would like to emphasize what we think is novel and interesting in our findings. Although we have not yet presented an exact mechanism for mTALT amplification, we still think it is a very interesting phenomenon that the mTALT sequence has an inherent ability to protect telomeres. There is a similarity to the situation in which the SV40 sequence is inserted into telomere and replicated, but there is a difference in that the sequence existing in the genome, rather than an external sequence, is contributing to the protection of the telomere. In any case, by showing that both human and mouse cells can stably maintain telomeres while harboring non-telomeric sequences, we successfully and meaningfully expanded our knowledge on the mammalian telomere physiology.

Following the reviewer's suggestions, we have changed the text in the second revision as follows.

"Our findings suggest that the robust telomere system based on simple repeats can convert to the ALT mechanism making use of non-telomeric sequences even in mammalian cells."

→ **Our findings suggest that non-telomeric sequences from an internal genomic region can be parts of telomeres even in mammalian cells.**

"The evolutionary conservation of this ALT mechanism implies that cells have a backup telomere maintenance mechanism capable of producing new telomeres."

→ **The evolutionary conservation of this ALT phenomenon implies that eukaryotes have a robust system to cope with the loss or inactivation of telomerase.**

"We found that a specific subtelomeric element, the mouse template for ALT (mTALT), is used for repairing telomeric DNA damage as well as for the development of new telomeric sequences in ALT-dependent mouse

embryonic stem cells.”

→ We found that a specific subtelomeric element, the mouse template for ALT (mTALT), is used for repairing telomeric DNA damage as well as for composing portions of telomeres in ALT-dependent mouse embryonic stem cells.

Q: I also have a problem with the novelty. First, this study is in mice, in which the significance, utility, mechanistic characterization and relevance of ALT is not clear.

A: Thank you for your thoughtful discussion. We agree with the reviewer that there have not been many studies on the ALT mechanism with mouse model. In addition, the fact that the average telomere length of the widely used reference mice model is ten times longer than that of humans suggests that it is difficult to apply the mouse model directly to human biology. However, despite these limitations, the mouse studies significantly contributed to the telomere biology as a whole. First of all, most of the telomere-binding proteins have been discovered through mouse cell studies, and these proteins are well preserved in humans. Considering the versatility of the telomere protection mechanism, the mouse model may be considered to be an important axis of mammalian telomere research including ALT. The problem has been rather that there has not been an adequate mouse model to study ALT physiology. Our telomerase-deficient mESC model can be proven as a model for longitudinal studies on ALT initiation and maintenance in mammalian cells. Second, the mouse was the first animal model that showed that the recombination-based telomere maintenance mechanism could have a physiological meaning even in the presence of telomerase. Considering that most ALT mechanisms are based on recombination, it is reasonable that a mechanism similar to ALT may be important in the mouse development. In addition, when telomerase-based cancer was induced in mice and a telomerase inhibition strategy was introduced, cancers with the ALT mechanism occurred. Although it is limited to certain conditions, it is true that ALT can function in mice.

ALT mechanisms have been studied in terms of the mechanism of telomere maintenance in cancers, but studies on the functions of these mechanisms in real individuals have been limited. Although we have not yet been able to present a specific mechanism for telomere maintenance in the mESC study, ALT mESC can be an important model to investigate ALT mechanisms regarding the gold standard definition of ALT that the ability to maintain telomere length without telomerase. We can infer that the loss of the telomerase gene is not a necessary prerequisite for the mTALT sequence to be replicated because we observed in some wild mice strains that the mTALT sequence was replicated in natural conditions with functional telomerase present. If we use our model to clarify the activation process of this mechanism, it could contribute to a new understanding of the stress and physiological situations in which ALT can function in mammals *in vivo*.

Reflecting these considerations on the pros and cons of the mouse embryonic stem cell model for telomere biology including ALT, we added the following paragraph to the discussion section of the revised

manuscript.

“Although there have been reports that the ALT-like mechanism is involved in telomere maintenance during mouse development, the mouse organismal model that maintains telomeres only by ALT has not been established. Considering that telomeres of mice are considerably longer than that of humans, it is still unclear whether the mechanism of telomere maintenance found in mice can be applied universally to various mammals including humans. In addition, further studies are needed to determine whether the ALT phenomenon identified in this study works in the same way as in human ALT cancer cells in which mechanisms based on homologous recombination or break-induced replication operate.”

“Our report is the first in mammalian models to longitudinally track and describe the telomeric changes integrating the non-telomeric sequence existing in the genome. Considering that this phenomenon is widely conserved in various model organisms such as *S. cerevisiae*, *S. pombe*, *C. elegans* and specific human immortalized cell line (AG11395), the characteristics of ALT revealed in this study will contribute to the expansion of our knowledge on telomere biology”

In addition, we changed the title and the sentence in abstract to clearly show our study focuses on the mouse embryonic stem cell model. The title was changed from “Mammalian telomeres reformed with non-telomeric sequences” to “Telomeres reformed with non-telomeric sequences in mouse embryonic stem cells.

In abstract, “Here we show that mammalian telomeres can also exploit non-telomeric, unique sequences in addition to telomeric repeats” was changed to:

→ “Here we show that mouse telomeres can exploit non-telomeric, unique sequences in addition to telomeric repeats.”

Q: Second, a similar process has been observed in human ALT cells (as mentioned in my previous review). Although, these back to back studies have now been mentioned in the Discussion, they are not discussed fully or described in the Introduction, which I feel is important to set the scene for this work.

A : In Introduction, we added the description of AG11395 and explained that APB was not found, thus introducing the possibility that ALT may not be all the same mechanisms.

“While telomeric repeats and unique sequences are used to maintain telomeres in these various species, telomeres of human ALT cancers seem to consist only of telomeric repeats and their variants.”

→ “Telomeres of human ALT cancers seem to consist only of telomeric repeats and their variants. Interestingly, there is a distinct human cell line, AG11395, which is a SV40-transformed Werner mutant fibroblast. Telomeres of AG11395 cells contain extensive amounts of SV40 sequences and telomeric

repeats. The cells showed some of typical ALT characteristics, but lacked ALT-associated promyelocytic leukaemia bodies (APBs), a molecular marker of ALT. This case implies the possibility that ALT may be a multi-faceted mechanism.”

Q: In the abstract, please state ALT as alternative lengthening of telomeres (not alternative telomere lengthening), telomere loss does not directly induce telomere recombination.

A: We changed “alternative telomere lengthening” to “alternative lengthening of telomeres”. We also revised the following sentence.

“In ALT, telomere loss can induce telomere recombination by which specific sequences can be recruited into telomeres; however, to date, only canonical telomeric repeat-based telomeres have been found in mammals.”

→ “ALT mainly utilizes recombination-based replication mechanisms and the constituents of ALT-based telomeres vary depending on models.”

Q: In the Introduction, please correct the first sentence, change the phrase "telomeres can be "broken" by cellular stresses".

A: We have revised the sentence as follows.

“All eukaryotic cells have linear chromosomes which inevitably cause the end replication and the end protection problem.”

→ “Ends of linear chromosomes should handle two problems: ‘the end replication problem’ in which DNA replication machinery cannot completely replicate ends of lagging strands and ‘the end protection problem’ in which chromosomal ends should be discriminated from internal double-strand breaks (DSBs).

We also changed “broken” to “damaged”.

Q: I'm also not convinced by the telomere analysis in Fig 1, for instance in Fig 1j are we not supposed to be seeing TTAGGG repeats being replaced by mTALT from PD100 to PD800? I cannot see this in the figure. Also, have these cell lines been STR profiled for authenticity? I can't find this in the methods. The chromosomes look very different at PD100 compared to PD800.

A: The FISH results clearly showed that the pattern shown in the results of the WGS analysis (Fig. 1c, 1h) and the results of mmqPCR (Fig. 1i) are the phenomena occurring at the actual ends, not at interstitial repeats. Precisely speaking, the telomere length decrease occurs only when proceeding from PD100 to PD350. Although there was a change in copy number in WGS, mTALT amplification was not detected in FISH due to too small amount of mTALTs (PD100 : ~2 copy, PD350 : ~4 copy). When proceeding from PD350 to PD450, the recovery of TTAGGG and mTALT recruitment occur simultaneously. In PD450,

mTALT is not located at the entire ends and shows partial localization. And from PD450 to PD800, mTALT spreads to the entire ends and coexist with telomere repeats (yellow signal). To show this a little more clearly, additional images were taken and replaced the previous figures.

When it comes to morphology of chromosome, as shown in the previous paper (Niida et al., 2000, Mol Cell Biol.), PD100 cell had few chromosomal fusions, and after PD350, the number of chromosomes significantly decreased due to telomere-shortening-induced fusions. Mouse chromosomes are acrocentric, and they can behave like stable metacentric chromosomes when fusions occurred between short-arm telomeres. The chromosomes look differently because of the chromosomal fusions that occurred more frequently in later cells.

Additionally, SNP profiling by whole genome sequencing (WGS) is much more accurate than STR typing for cell line authentication, and our data are the case. In particular, SNP profiling-based authentication is an appropriate method for mouse cell lines from inbred strains and the cell lines with high genome instability. According to Yu et al. Nature (2015) titled as “A resource for cell line authentication, annotation and quality control”, $\geq 90\%$ identity score by SNP profiling was recommended. The identity score using about 2.3 million SNPs for PD100 and PD800 cell lines was 94.28%, indicating that the WGS data ensure the authenticity. The reviewer might think that some chromosomes looked like human chromosomes. When the data were mapped onto the human reference genome, the rates were 23.44% and 24.28% for PD100 and PD800, respectively. Therefore, the cell lines were not contaminated

by other cell lines, and PD100 and PD800 originated from the same cell line.

REVIEWERS' COMMENTS

Reviewer #1 (Remarks to the Author):

I am happy with the changes the authors have made and I am supportive of publication.

I have a few minor edits:

- (i) line 102 "showed some typical" (remove "of")
- (ii) line 122 "from the genome"
- (iii) line 473 "than those of humans"
- (iv) line 478 edit the first sentence (unclear)
- (v) line 481 "and the specific"

Responses to reviewer's comments

Reviewer #1 (Remarks to the Author):

Q: have a few minor edits:

(i) line 102 "showed some typical" (remove "of")

(ii) line 122 "from the genome"

(iii) line 473 "than those of humans"

(iv) line 478 edit the first sentence (unclear)

(v) line 481 "and the specific"

A: We have performed the edits (i), (ii), (iii), and (v) as suggested by the reviewer. For (iv), we edited the sentence as follows:

“Our report is the first in mammalian models to longitudinally track and describe the telomeric changes integrating the non-telomeric sequence existing in the genome.”

→ “Our report is the first in mammalian models to longitudinally analyse telomeres maintained with non-telomeric sequences.”

All changes are marked as red colour.

Thank you.